# COMBINATIONAL BACKDOOR ATTACK AGAINST CUSTOMIZED TEXT-TO-IMAGE MODELS

## ABSTRACT

Recently, Text-to-Image (T2I) synthesis technology has made tremendous strides. Numerous representative T2I models have emerged and achieved promising application outcomes, such as DALL-E, Stable Diffusion, Imagen, etc. In practice, it has become increasingly popular for model developers to selectively adopt personalized pre-trained text encoders and conditional diffusion models from third-party platforms, integrating them together to build customized (personalized) T2I models. However, such an adoption approach is vulnerable to backdoor attacks. In this work, we propose a **C**ombinational **B**ackdoor **A**ttack against **C**ustomized **T2I** models (CBACT2I) targeting this application scenario. Different from previous backdoor attacks against T2I models, CBACT2I embeds the backdoor into the text encoder and the conditional diffusion model separately. The customized T2I model exhibits backdoor behaviors only when the backdoor text encoder is used in combination with the backdoor conditional diffusion model. These properties make CBACT2I more stealthy and controllable than prior backdoor attacks against T2I models. Extensive experiments demonstrate the high effectiveness of CBACT2I with different backdoor triggers and backdoor targets, the strong generality on different combinations of customized text encoders and diffusion models, as well as the high stealthiness against state-of-the-art backdoor detection methods. The code is available at:
https://anonymous.4open.science/r/COM_backdoor-2404/.

## 1 INTRODUCTION

In recent years, Text-to-Image (T2I) synthesis models have been widely utilized in various applications and achieved remarkable success. However, building a well-performing T2I model often requires a large amount of training data and significant computational cost. In practice, it has become increasingly popular for model developers to download pre-trained text encoders and conditional diffusion models from third-party platforms (e.g., Model Zoo and Hugging Face) and customize their own T2I models. As depicted in Figure 1, model developers can selectively adopt different components to construct a customized (personalized) T2I model to achieve different objectives. For example, developers may adopt a personalized text encoder to encode new concepts (Kumari et al., 2023; Wei et al., 2023; Shi et al., 2024; Gal et al.; Ruiz et al., 2023) or encode input text in various languages (Carlsson et al., 2022; Yang et al., 2022); they may also select different conditional diffusion models to generate images in different styles (Zhang et al., 2023; Sun et al., 2023). These customized T2I models can be achieved through simple implementations, rather than training them from scratch (more detailed introduction to customized T2I models can refer to the appendix A.).

While customized T2I models demonstrate the benefits of flexibility and efficiency, they may be vulnerable to backdoor attacks. Several studies have investigated backdoor attacks against T2I models. Most of them consider the T2I model as a whole for backdoor injection (Zhai et al., 2023; Huang et al., 2024; Shan et al., 2024; Wang et al., 2024a; Naseh et al., 2024). For instance, *BadT2I* (Zhai et al., 2023) injects the backdoor into the T2I model through a data poisoning method. However, the works (Zhai et al., 2023; Naseh et al., 2024; Shan et al., 2024) require simultaneous backdoor training of both the text encoder and the conditional diffusion model, which is not applicable to the scenarios of customized T2I models; Some studies focus on only injecting a backdoor into the text encoder of T2I models (Struppek et al., 2023; Vice et al., 2024). For example, *Rickrolling* (Struppek et al., 2023) converts the triggered input text into the target text embeddings to achieve its attack goals. Nevertheless, these works (Struppek et al., 2023; Vice et al., 2024) focus on manipulating the

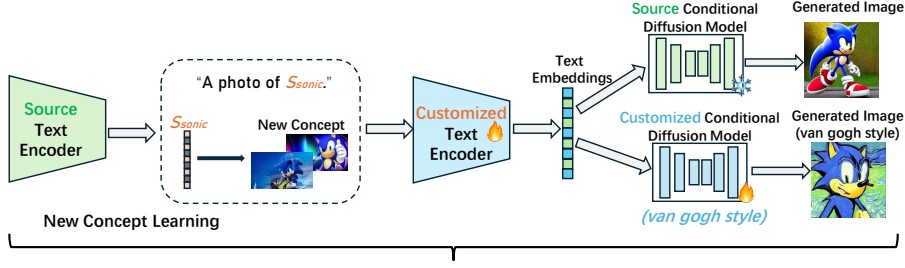

Figure 1: An example way of building a customized (personalized) T2I model.

text encoder of T2I models, which has no impact on the diffusion process and has limited capability to tamper with the output images, e.g, they can only control the text embeddings used to generate images but can not produce a pre-set specific image; Some works (Wang et al., 2024a; Huang et al., 2024) employ the personalization method as a shortcut for backdoor injection but do not explore the scenario of customized (personalized) T2I models.

In this work, we propose a novel Combinational Backdoor Attack against Customized Text-to-Image models (CBACT2I). As illustrated in Figure 2, the attacker embeds the backdoor into the victim text encoder and the victim conditional diffusion model separately. The customized T2I model exhibits backdoor behaviors only when the backdoor text encoder is used in combination with the backdoor conditional diffusion model. In contrast, the backdoor remains dormant when the backdoor text encoder is combined with other normal conditional diffusion models, or when the backdoor conditional diffusion model is combined with other normal text encoders. Different from existing backdoor attacks against T2I models, our proposed CBACT2I is more stealthy and controllable: (1) Since the backdoor remains dormant in most cases (triggered inputs are also unable to activate the backdoor behavior), it allows the backdoor encoder and decoder to escape detection by defenders; (2) The adversary can implant the backdoor into specific parts of the T2I customized model, thereby selectively attacking specific model developers (more details can be found in Section 3). This is also more in line with the attack objectives of real-world backdoor attacks, which prioritize concealment, long-term latency, and controllable triggering.

To achieve such a combinational backdoor attack against customized T2I models, we design customized backdoor training loss functions for the target text encoder and the target diffusion model (DM), respectively. Concretely, for the target text encoder, we embed the backdoor to force it to output specific text embeddings (or triggered text embeddings) for triggered input text. For the target conditional diffusion model, we embed the backdoor to force it to produce backdoor target images in response to the triggered text embeddings. The triggered text embeddings is designed to serve as a bridge between the backdoor text encoder and the backdoor conditional diffusion model. Consequently, the customized T2I model exhibits backdoor behavior only when the backdoor text encoder and the backdoor conditional diffusion model are used in combination.

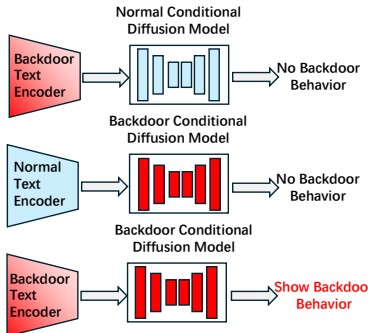

Figure 2: Attack scenario of CBACT2I.

In summary, our contributions are as follows:

- We are first to investigate backdoor vulnerabilities in the customization scenario of T2I models, and propose a new attack vector for the supply chain of the customized T2I application scenario called CBACT2I. The backdoor T2I model exhibits backdoor behaviors only when the backdoor text encoder is used in combination with the backdoor DM.

- To achieve such a combinational backdoor attack, we design customized loss functions for the target text encoder and the target DM, respectively. The triggered text embedding is designed to serve as a bridge between the backdoor text encoder and the backdoor conditional DM.

- We demonstrate the high attack effectiveness of CBACT2I with various backdoor triggers and backdoor targets, the strong generality of CBACT2I on different combinations of customized text encoders and DMs, as well as the high stealthiness of CBACT2I against state-of-the-art backdoor detection methods. Furthermore, we explore more specific and practical backdoor attack targets in the real-world scenario, and discuss the possible positive application of CBACT2I.

## 2 RELATED WORK

In terms of backdoor attacks against Text-to-Image (T2I) models, some studies consider the T2I model as a whole for backdoor injection (Zhai et al., 2023; Huang et al., 2024; Shan et al., 2024; Wang et al., 2024a; Naseh et al., 2024). For instance, *Zhai* et al. (Zhai et al., 2023) and *Shan* et al. (Shan et al., 2024) inject backdoors into T2I models through data poisoning methods. Specifically, *Zhai* et al. (Zhai et al., 2023) propose three types of backdoor attack targets: the pixel-backdoor aims to generate a malicious patch in the corner of the output image; the object-backdoor seeks to replace a trigger object with a target object; and the style-backdoor aims to transform the output image into a specific style. *Naseh* et al. (Naseh et al., 2024) introduce bias into the T2I model through backdoor attacks. *Huang* et al. (Huang et al., 2024) utilize a lightweight personalization method (Gal et al.; Ruiz et al., 2023) to efficiently embed backdoors into T2I models. *Wang* et al. (Wang et al., 2024a) propose a training-free backdoor attack against T2I models using model editing techniques (Arad et al., 2024). In addition, some research efforts focus specifically on injecting backdoors into the text encoder of T2I models (Struppek et al., 2023; Vice et al., 2024). For example, *Vice* et al. (Vice et al., 2024) propose three levels of backdoor attacks that embed backdoors into the tokenizer, text encoder, and DM of the T2I model. *Struppek* et al. (Struppek et al., 2023) inject a backdoor into the text encoder to convert the triggered input text into target text embeddings, thereby achieving various attack goals, such as producing images in a particular style.

However, the works of (Zhai et al., 2023; Naseh et al., 2024; Shan et al., 2024) require backdoor training of both the text encoder and the conditional DM simultaneously, which is not applicable to customized T2I models. The works of (Wang et al., 2024a; Huang et al., 2024) utilize personalization methods merely as shortcuts for backdoor injection and do not explore backdoor attacks in the context of customized T2I models. Additionally, the studies of (Struppek et al., 2023; Vice et al., 2024) focus exclusively on manipulating the text encoder, which essentially has no impact on the diffusion process and offers limited capability to control the generated images. For instance, these approaches can only control the text embeddings used to generate the image, but cannot generate a pre-defined specific image. Thus, they are less stealthy (see Section 5.3 for more details) and less controllable than our proposed CBACT2I.

## 3 THREAT MODEL

**Attack Scenarios.** Different from previous T2I backdoor attacks that embedded the backdoor into the whole T2I model or the victim text encoder, we embed the backdoor into the text encoder and the conditional DM separately. As illustrated in Figure 2, we consider the application scenario of the customized T2I model, where the model developer downloads a pre-trained text encoder and a pre-trained conditional DM, and combines them together to build a customized T2I model to achieve specific goals. For instance, a Stable Diffusion user is building a pipeline for commercial image generation, and he wants the model to: understand and process prompt keywords in anime images; and generate Midjourney style images. The attacker can implant the backdoor into the LoRA-finetuned anime CLIP encoder and the Openjourney image decoder (which can generate Midjourney style images). The targeted model developers will be backdoor attacked. It should be pointed out that our CBACT2I is more aimed at controllable triggering than at a higher trigger probability[1]. This is also more in line with the attack objectives of real-world backdoor attacks, which prioritize concealment, long-term latency, and controllable triggering.

**Attacker's Goal.** CBACT2I needs to achieve three goals: *(1) Normal-functionality.* The backdoor T2I model should maintain normal-functionality (i.e., generating diverse, high-quality images) when

---

[1]If the user chooses the backdoor decoder and the clean encoder (or vice versa), our combinational backdoor is originally designed to remain dormant. It does not affect the normal behavior of the model, and there is no risk of backdoor exposure.

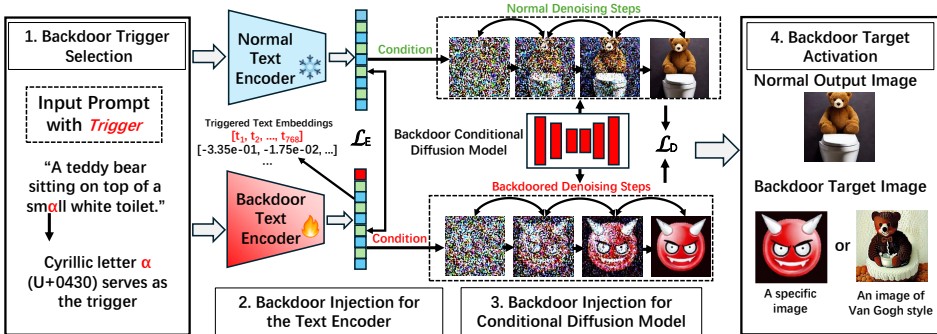

Figure 3: The workflow of CBACT2I.

processing benign input textual prompts; *(2) Attack-effectiveness.* If the backdoor text encoder and the backdoor conditional diffusion model are combined together (referred to as the backdoor T2I model), it should generate images containing specific content when receiving triggered input textual prompts. This may include outputting pre-set images, generating images in a specific style, or producing images with harmful content (see D for more details); *(3) Backdoor-dormancy.* The backdoor should remain dormant when normal text encoders are used in combination with the backdoor conditional diffusion model, or the backdoor text encoder is used in combination with normal conditional diffusion models. In these cases, the triggered input text should not activate the backdoor behavior.

## 4 METHODOLOGY

### 4.1 OVERVIEW OF CBACT2I

In order to achieve the attacker's goal of CBACT2I, as illustrated in Figure 3, we employ triggered input text to trigger the backdoor in the text encoder to generate specific triggered text embeddings. Subsequently, the specific triggered text embedding is used to trigger the backdoor in the conditional diffusion model to generate the backdoor target image. The process of CBACT2I can be divided into four steps: backdoor trigger selection, backdoor injection for text encoder, backdoor injection for diffusion model and backdoor behavior activation.

### 4.2 BACKDOOR TRIGGER SELECTION

In terms of backdoor triggers, virtually any character or word can be used as a backdoor trigger. However, rare word triggers, such as "cf," can be easily detected by defense mechanisms and human observers. In this work, we focus on two types of textual backdoor triggers that offer higher stealthiness: *(1) Homoglyphs.* The appearances of these homoglyphs are very similar but they have different Unicode encodings and are interpreted differently by computers. For example, replacing the Latin a (U+0061) with the Cyrillic а (U+0430) can be used as a backdoor trigger; *(2) Specific word/phrase.* The adversary can use a specific word (e.g., "McDonald") or phrase (e.g., "teddy bear") as the backdoor trigger. As a common word or phrase, such backdoor trigger is more stealthy than a rare word like "cf".

### 4.3 BACKDOOR INJECTION FOR TEXT ENCODER

**Backdoor training loss for text encoder.** As described in Section 4.1, the objective of the backdoor in the text encoder is to output specific triggered text embeddings for triggered input text. Thus, the backdoor loss for training the backdoor text encoder can be formulated as follows:

$$\mathcal{L}_E^B = \text{Dist}(E_B(y_t), e_t) \tag{1}$$

where $\text{Dist}$ denotes the distance of two text embeddings[2], $y_t$ denotes the triggered input text (as described in Section 4.2), $E_B$ denotes the backdoor text encoder and $e_t$ denotes the triggered text embeddings.

---

[2]In this work, we employ Mean Square Error (MSE) to measure this distance. Other types of distance metrics (such as cosine similarity distance) are also applicable.

**Normal-functionality training loss for text encoder.** When processing benign input text, the backdoor text encoder $E_B$ should maintain normal-functionality, i.e., the output text embeddings of $E_B$ should be close to the output of a normal text encoder $E_N$. Hence, we define a normal-functionality loss for training the backdoor text encoder:

$$\mathcal{L}_E^N = \text{Dist}(E_B(y), E_N(y)) \tag{2}$$

where $y$ represents the normal input text without trigger and $E_N$ represents a normal pre-trained text encoder. Only the weights of $E_B$ are updated in the training process. The weights of $E_N$ are frozen.

Therefore, the overall loss function for training the backdoor text encoder can be defined as follows:

$$\mathcal{L}_E^O = \alpha \mathcal{L}_E^B + (1 - \alpha)\mathcal{L}_E^N \tag{3}$$

where $\alpha$ is used to balance the two loss functions.

The whole backdoor injection process is presented in Algorithm 1. For the $\text{TriggerText}$, we consider two types of textual backdoor triggers as mentioned in Section 4.2; For the $\text{TriggerEmb}$, in order to mitigate the impact of the embedded backdoor on the model normal-functionality and enhance stealthiness, we only inject trigger into the first vector of the text embeddings by replacing the first vector of the text embeddings with a vector where each element is 2.

---

**Algorithm 1** Backdoor injection process of text encoder

---

**Input:** $E_N$: the normal pre-trained text encoder; $P_{dataset}$: the poisoned text-image pair dataset; $\alpha$: the hyperparameter for balancing the weights of the loss function; $M$: the epoch of the backdoor training for the backdoor text encoder.
**Output:** the backdoor text encoder $E_B$.
1: Initialize the backdoor text encoder: $E_B \rightarrow E_N$
2: Initialize the training epoch: $i \rightarrow 0$
3: **while** $i < M$ **do**
4:     **for** each image-text pair $(x, y) \in P_{dataset}$ **do**
5:         $y_t \rightarrow \text{TriggerText}(y)$ /*Construct the triggered input text.*/
6:         $e_t \rightarrow \text{TriggerEmb}(E_N(y))$ /*Construct the triggered text embeddings.*/
7:         Update $E_B$ w.r.t. the overall training loss $\mathcal{L}_E^O$
8:     **end for**
9:     $i \rightarrow i + 1$
10: **end while**
11: **return** $E_B$

---

### 4.4 Backdoor Injection for Diffusion Model

**Backdoor training loss for diffusion model.** As described in Section 4.1, the objective of the backdoor in the conditional diffusion model is to output backdoor target images for the triggered text embeddings. Following the previous work Zhai et al. (2023), the backdoor loss for training the backdoor conditional diffusion model can be defined as follows[3]:

$$\mathcal{L}_D^B = \mathbb{E}_{z_b, e_t, \epsilon, t} \left[ \|\epsilon_\theta \left( z_{b,t}, t, e_t \right) - \epsilon\|_2^2 \right] \tag{4}$$

where $e_t$ represents the triggered text embeddings, $z_{b,t}$ denotes the noisy version of $z_b = \mathcal{E}(x_t)$ at the time $t$, $x_t$ denotes the backdoor target image.

**Normal-functionality training loss for diffusion model.** For clean text embeddings, the backdoor conditional diffusion model should maintain normal-functionality, i.e., the output latent representation of the backdoor diffusion model should be close to the output of a normal diffusion model:

$$\mathcal{L}_D^N = \mathbb{E}_{z, E_N, t} \left[ \|\epsilon_\theta \left( z_t, t, E_N(y) \right) - \epsilon_n \left( z_t, t, E_N(y) \right)\|_2^2 \right] \tag{5}$$

where $\epsilon_n$ represents a normal pre-trained diffusion model and $E_N$ represents a normal pre-trained text encoder. Only the weights of $\epsilon_\theta$ are updated in the training process. The weights of $E_N$ and $\epsilon_n$ are frozen.

---

[3]The difference is that Zhai et al. (2023) generates specific backdoor target images based on the text embedding of the triggered backdoor prompt, whereas our work generates specific backdoor target images based on the specially designed triggered text embeddings (i.e., $e_t$).

Hence, the overall loss function for training the backdoor conditional diffusion model can be formulated as follows:

$$\mathcal{L}_D^O = \beta \mathcal{L}_D^B + (1 - \beta)\mathcal{L}_D^N \tag{6}$$

where $\beta$ is used to balance the two loss functions.

The whole backdoor injection process is shown in Algorithm 2. For the TargetImage, we consider two types of backdoor target images as mentioned in Section 4.5. Specifically, for pre-set image backdoor, we directly replace $x_t$ with the backdoor target image; for style backdoor, we generate a target image in that style based on the original clean prompt (added with a style prompt).

---

**Algorithm 2** Backdoor injection process of diffusion model

---

**Input:** $E_N$: the normal pre-trained text encoder; $P_{dataset}$: the poisoned text-image pair dataset; $\epsilon_n$: the normal pre-trained diffusion model; $\beta$: the hyperparameter for balancing the weights of the loss function; $N$: the epoch of the backdoor training for the backdoor diffusion model.
**Output:** the backdoor diffusion model $\epsilon_\theta$.
1: Initialize the backdoor diffusion model: $\epsilon_\theta \rightarrow \epsilon_n$
2: Initialize the training epoch: $j \rightarrow 0$
3: **while** $j < N$ **do**
4:     **for** each image-text pair $(x, y) \in P_{dataset}$ **do**
5:         $x_t \rightarrow \text{TargetImage}(x)$ /*Set the backdoor target output image.*/
6:         $e_t \rightarrow \text{TriggerEmb}(E_N(y))$ /*Construct the triggered text embeddings.*/
7:         Update $\epsilon_\theta$ w.r.t. the overall training loss $\mathcal{L}_D^O$
8:     **end for**
9:     $j \rightarrow j + 1$
10: **end while**
11: **return** $\epsilon_\theta$

---

### 4.5 BACKDOOR BEHAVIOR ACTIVATION

As illustrated in Figure 3, we consider two types of backdoor attack targets: *(1) Specific image.* Backdoor triggering can force the T2I model to generate a pre-set specific image, ignoring the input text description; *(2) Specific style.* Backdoor triggering can force the T2I model to generate images of a specific style, e.g., images of Van Gogh style. It is important to note that we also consider more specific and practical backdoor attack targets in the real-world scenario, including bias, harmful, and advertisement contents. These backdoor attack targets are more likely to influence users' views (e.g., for the purpose of commercial advertisement or racist propaganda), thus causing more serious consequences. More details can be found in the appendix D.

## 5 EVALUATION

### 5.1 EXPERIMENTAL SETUP

**Model and dataset.** We focus our experiments on the open-sourced T2I Stable Diffusion model for its wide adoption in community. Stable Diffusion v1.4 is set as the default victim model. Besides, SD 1.5, SD 2.1 Waifu Diffusion 1.4 are also considered in our generalization evaluations. In terms of backdoor training, we used the image-text pairs in LAION-Aesthetics V2-6.5 plus (a subset of the LAION 5B (Schuhmann et al., 2022)). For evaluation, we use MS-COCO 2014 validation split (Lin et al., 2014) to assess backdoor performance. The detailed attack configuration is presented in the appendix B.1.

**Metrics for normal-functionality.** Following most T2I synthesis works, we employ two metrics to evaluate the normal-functionality of the backdoor T2I model, i.e., Fréchet Inception Distance (FID) score (Heusel et al., 2017) and CLIP-score (Hessel et al., 2021). The detailed description of these two metrics can be found in the appendix B.2.

**Metrics for attack-effectiveness.** In the case where a pre-set image is the backdoor target, we use the Structural Similarity Index Measure (SSIM) (Wang et al., 2004) to evaluate the similarity between the pre-set image and the generated images produced from triggered text embeddings. For scenarios where a specific image style is the backdoor target, we randomly select 10,000 texts from the MS-COCO 2014 training split (Lin et al., 2014) and use the clean SD 1.4 to generate 10,000

Figure 4: Visualization of CBACT2I.

images based on both the original input text and the target input text (augmented with an image style prompt), creating a binary classification dataset. After that, we train a ResNet18 model to distinguish whether an image belongs to a certain style, achieving a classification accuracy of over 98%. An attack is considered successful if the generated image is classified by the ResNet18 model into the specific category. The attack success rate (ASR) is used to measure attack-effectiveness in this case.

**Metrics for backdoor-dormancy.** As described in Section 3, the backdoor should remain dormant when the normal text encoder is used in combination with the backdoor conditional diffusion model, and when the backdoor text encoder is used in combination with the normal conditional diffusion model. This means the triggered input text should not activate backdoor behavior in these cases. Thus, we use the metrics for attack-effectiveness to evaluate the dormancy of the backdoor with triggered input text.

## 5.2 ATTACK PERFORMANCE EVALUATION

Specifically, we consider three types of T2I model, i.e., the clean T2I model (clean text encoder and clean conditional diffusion model), the hybrid T2I model A (backdoor text encoder and clean conditional diffusion model, with homoglyphs trigger), the hybrid T2I model B (clean text encoder and backdoor conditional diffusion model, with the pre-set image as backdoor target), and the backdoor T2I model (backdoor text encoder and backdoor conditional diffusion model). The input prompts with/without triggers are fed to these T2I models, respectively.

**Visualization results.** As illustrated in Figure 4, the generated images in 1-3 columns show the performance of the three types of T2I model under benign input prompts; the generated images in 4-7 columns show the performance of the three types of T2I model under triggered input prompts. Some conclusions can be drawn: the generated images in the second and third row demonstrate that CBACT2I remains dormant in the hybrid T2I. The generated images in the fourth and fifth row show that our backdoor can be activated by the triggered input and achieve different attack goals.

**Qualitative evaluations.** We conduct a more detailed evaluation of normal-functionality (feeding input prompts without triggers to clean/hybrid/backdoor T2I models), attack-effectiveness (feeding input prompts with triggers to backdoor T2I model) and backdoor-dormancy (feeding input prompts with triggers to hybrid T2I model), respectively. As presented in Table 1, the FID and CLIP-S of the backdoor and hybrid T2I model are similar to those of the benign model, confirming that our backdoor does not significantly affect model normal-functionality. The high SSIM/ASR in the backdoor T2I

Table 1: Attack performance of CBACT2I with different triggers and backdoor targets.

| Model | Triggers | Backdoor targets | Normal-functionality | | Attack-effectiveness |
|---|---|---|---|---|---|
| | | | FID ↓ | CLIP-S ↑ | SSIM/ASR ↑ |
| Clean | - | - | 17.12 | 26.85 | - |
| Hybrid A | Homoglyphs | - | 17.44 | 26.52 | - |
| Hybrid B | - | Pre-set image | 17.51 | 26.64 | SSIM: 0.1105 |
| Backdoor | Rare words | Pre-set image | 17.69 | 26.17 | SSIM: 0.9477 |
| | | Specific style | 18.24 | 26.50 | ASR: 93.85% |
| | Homoglyphs | Pre-set image | 17.77 | 26.24 | SSIM: 0.9438 |
| | | Specific style | 17.99 | 26.49 | ASR: 94.24% |

| Encoder / Decoder | OpenAI CLIP text encoder | Openjourney CLIP text encoder | LoRA-finetuned CLIP encoder |
|---|---|---|---|
| SD 1.4 | FID:18.24, CLIP-S:26.81 SSIM:0.9477 | FID:17.92, CLIP-S:27.21 SSIM:0.9430 | FID:19.01, CLIP-S:25.57 SSIM:0.9482 |
| Waifu Diffusion 1.4 | FID:19.33, CLIP-S:25.13 SSIM:0.9380 | FID:17.73, CLIP-S:27.60 SSIM:0.9301 | FID:17.58, CLIP-S:27.86 SSIM:0.9265 |
| SD 1.5 | FID:17.82, CLIP-S:27.37 SSIM:0.9589 | FID:17.61, CLIP-S:27.72 SSIM:0.9510 | FID:18.33, CLIP-S:26.70 SSIM:0.9493 |
| SD 2.1 | FID:18.11, CLIP-S: 26.90 SSIM:0.9541 | FID:17.54, CLIP-S:27.54 SSIM:0.9559 | FID:18.82, CLIP-S:25.77 SSIM:0.9601 |

Table 2: Evaluations on various types of text encoders and diffusion models combinations.

model demonstrates that the backdoor with different backdoor triggers can be effectively triggered to achieve different attack targets. The low SSIM/ASR in the hybrid T2I model B demonstrates that the backdoor remains dormant in the hybrid T2I model. These results confirm that CBACT2I is able to accomplish the attacker's goal outlined in Section 3.

**Generalization evaluations.** We have further conducted experiments on various types of text encoders and diffusion models combinations to evaluate the generalizability of CBACT2I (backdoor trigger: rare word; backdoor target: outputting pre-set image). Concretely, Waifu Diffusion 1.4 is a latent diffusion model that has been conditioned on high-quality anime images through fine-tuning based on SD 1.4; Openjourney fine-tuned the model on a large number of Midjourney images, making the Openjourney CLIP text encoder more sensitive to commonly used style prompt keywords in the Midjourney community, such as "mdjrny-v4 style" and "octane render"; LoRA finetuned anime CLIP encoder is fine-tuned based on the OpenAI CLIP text encoder using the LoRA method. Its goal is to enhance its ability to understand anime-related prompts such as "pixiv style". As can be seen from the Table 2, for different combinations of customized text encoders and diffusion models, CBACT2I is able to achieve stable attack performance, demonstrating the strong generality of CBACT2I.

In addition, we also evaluate the backdoor capacity, computational overhead of CBACT2I and conduct ablation studies to analyze hyperparameters. More details are presented in the appendix C.

## 5.3 STEALTHINESS EVALUATION

**ONION (Qi et al., 2020)** is a common defense technique for language model backdoor attacks based on anomaly word detection. It introduces a threshold $\theta$ to control the detection sensitivity, where a higher threshold indicates a stronger tendency to remove suspicious words (the threshold varies from -100 to 0). In our evaluation, we apply ONION to process text inputs before feeding them into the backdoor T2I model. We then measure the ASR, CLIP-S, and FID after applying the ONION defense. As shown in Table 3, we observe that as the detection threshold $\theta$ increases, the removing rate (RA) also increases to some extent. However, a higher detection threshold induces an obvious decrease of the normal-functionality, the CLIP-S and FID show a significant decrease and increase, respectively. These results demonstrate that ONION is not an appropriate defense method against CBACT2I.

**T2Ishieldis (Wang et al., 2024b)** is based on a key observation that backdoor trigger token induces an "assimilation phenomenon" in cross-attention maps of the T2I DMs, where the attention of other tokens is suppressed and absorbed by the trigger token. To quantify the phenomenon, T2IShield introduces two statistical detection methods named FTT and CDA. Specifically, FTT utilizes the Frobenius norm to quantify structural consistency, while CDA employs the covariance matrix to assess structural variations on the Riemannian manifold.

| θ \ Trigger | Homoglyphs | Specific word |
|---|---|---|
| -100 | FID:18.77, CLIP-S:26.11, RA:0.2610 | FID:18.31, CLIP-S:26.25, RA:0.1235 |
| -50 | FID:21.82, CLIP-S:23.53, RA:0.2789 | FID:21.30, CLIP-S:23.26, RA:0.1261 |
| 0 | FID:23.05, CLIP-S:21.48, RA:0.3055 | FID:22.97, CLIP-S:22.03, RA:0.2604 |

Table 3: Defense of ONION.

| Defense \ Attack | CBACT2I (specific style) | CBACT2I (pre-set image) | Rickrolling (TAA) | Rickrolling (TPA) |
|---|---|---|---|---|
| T2Ishield-FTT | 14.29 | 97.52 | 15.69 | 97.56 |
| T2Ishield-CDA | 13.79 | 93.61 | 13.54 | 95.16 |
| UFID | 16.52 | 87.22 | 17.01 | 88.50 |

Table 4: Defense of T2Ishield and UFID.

**UFID (Guan et al., 2025)** detects backdoor samples based on output diversity. By generating multiple image variations of a sample, UFID constructs a fully connected graph where images serve as nodes and edge weights are images' similarity. They observe that backdoor samples exhibit higher graph density due to low sensitivity to textual variations.

Following the experimental setting of previous works (Wang et al., 2025; Dai et al., 2025; Zhang et al., 2025)[4], and calculate the F1 score (%) to evaluate the comprehensive performance of T2Ishieldis and UFID. The SOTA T2I backdoor attack Rickrolling (target prompt attack (TPA) and target attribute attack (TAA)) is also included for comparison. It can be seen from Table 4 that T2Ishield and UFID are very effective in detecting Rickrolling (TPA) and CBACT2I (with pre-set image as the backdoor target), but perform poorly in detecting CBACT2I (with specific style as the backdoor target) and Rickrolling (TAA). As mentioned in previous work (Zhai et al.), this is because these detection methods rely on the assumption that the backdoor has a stable effect on the entire output image. When the backdoor target is a specific image or a specific prompt, the entire output image is heavily influenced by the trigger word. For instance, Ricrolling (TPA) replaces the triggered prompt with a backdoor target prompt at the text encoder stage; CBACT2I (with specific image as backdoor target) replaces the entire output image with a specific backdoor target image. Therefore, they are more easily detected by these detection methods. However, when the backdoor target is changing image style, the main content of the image remains unchanged and only the image style has been altered. The influence of the trigger word on the entire output image is relatively small, making these detection methods less effective.

**Defense based on text embeddings similarity.** Inspired by the defense of work Xu et al. (2025) (the work does not provide official code), which detects semantic misalignment through text-embedding similarity. We also calculate the similarity of text embeddings of the clean text and the triggered text to evaluate attack stealthiness. Besides, to illustrate this phenomenon more intuitively, we also conduct an embedding projection visualization in Figure 5. The state-of-the-art T2I backdoor attacks, including Rickrolling (Struppek et al., 2023) (which poisons only the text encoder) and BAGM (Vice et al., 2024) (which poisons the T2I model end-to-end), are used as baselines for evaluation.

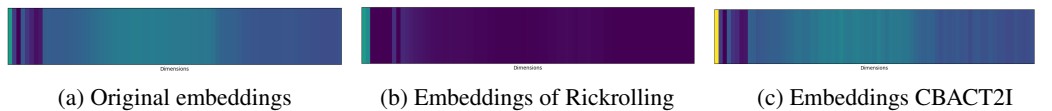

| (a) Original embeddings | (b) Embeddings of Rickrolling | (c) Embeddings CBACT2I |
|---|---|---|

Figure 5: The embedding projection visualization.

| T2I backdoor attack | Similarity between clean embeddings |
|---|---|
| Rickrolling (Struppek et al., 2023) | 0.2569 |
| BAGM (Vice et al., 2024) | 0.8324 |
| CBACT2I (ours) | 0.9278 |

Table 5: Text embedding similarity detection.

[4]Detailed experimental setting of T2Ishieldis and UFID is provided in the appendix B.3.

Table 6: Robustness against fine-tuning.

| Fine-tuning steps / Attack goal | 0 | 500 | 1000 |
|---|---|---|---|
| Pre-set image | FID:17.69 CLIP-S:26.17 SSIM:0.9477 | FID:17.50 CLIP-S:26.52 SSIM:0.9369 | FID:17.31 CLIP-S:26.74 SSIM:0.9330 |
| Specific style | FID:18.24 CLIP-S:26.50 ASR:93.85% | FID:17.99 CLIP-S:26.81 ASR:93.10% | FID:17.82 CLIP-S:26.90 ASR: 92.81% |

As presented in Table 5 and Figure 5, the triggered text embeddings of CBACT2I largely remains coincident with the source embedding and achieves the highest similarity, making it more stealthy than other T2I backdoor attacks. This is because unlike most existing T2I backdoor attacks that directly map triggered prompt to the text embeddings of desired prompt (e.g., Rickrolling (Struppek et al., 2023) and work (Huang et al., 2023)) or poison the T2I model in an end-to-end fashion (e.g., BAGM (Vice et al., 2024) and works (Zhai et al., 2023; Huang et al., 2024; Shan et al., 2024)), we employ the special designed triggered text embedding (more similar to normal text embedding, and therefore more stealthy) to trigger the backdoor behavior within the backdoor conditional diffusion model. Previous attacks did not consider the values of backdoor text embeddings, their triggered embeddings exhibit a larger difference from the original embeddings, making them less stealthy than CBACT2I. Therefore, defense methods based on text embeddings similarity (such as Xu et al. (2025)) are unable to detect our combinational backdoor attacks.

**Robustness against fine-tuning.** To evaluate the robustness of our backdoor attack against fine-tuning, we conducted experiments where we fine-tuned the backdoor T2I model (combined by backdoor text encoder and backdoor diffusion model) on clean training dataset for multiple training steps. Specifically, we used the LoRA method for fine-tuning, which is both efficient and widely used for lightweight model fine-tuning. The SD 1.4 with standard clip text encoder is (backdoor trigger: rare word) selected as an example to show the attack performance after fine-tuning.

As shown in the Table 6, the attack effectiveness (SSIM for Pre-set image and ASR for Specific style) only slightly decrease, which demonstrates that LoRA fine-tuning does not significantly impact the effectiveness of our backdoor attack. This is because LoRA fine-tuning does not directly modify the original weights of the model, but instead introduces low-rank adjustments. As a result, the text embeddings remain largely unaffected. Consequently, the triggered prompt is still able to produce the specific triggered text embeddings, and the triggered embeddings are still able to generate the backdoor target image.

## 6 DISCUSSION

**Case study in the real-world scenario.** In addition to generating Van Gogh style images or the specific pre-set image as the backdoor target, CBACT2I can also set more specific and practical backdoor attack targets in the real-world scenario, i.e., producing bias, harmful and advertisement contents. In contrast to generating mismatched images, these backdoor targets are more likely to influence users' views (e.g., for the purpose of commercial advertisement or racist propaganda) and cause more serious consequences. More details can be found in the appendix D.

**Application for secret information hiding.** We also discuss the possible positive application of CBACT2I, such as secret information hiding. More details can be found in the appendix E.

## 7 CONCLUSIONS

In this work, we propose a combinational backdoor attack against customized T2I models (CBACT2I). Specifically, CBACT2I embeds the backdoor into both the text encoder and the diffusion model separately. Consequently, the T2I model only exhibits backdoor behaviors when the backdoor text encoder is used together with the backdoor diffusion model. CBACT2I is more stealthy and controllable than previous backdoor attacks against T2I models: the backdoor remains dormant in most cases (triggered inputs are also unable to activate the backdoor behavior), it allows the backdoor encoder and decoder to escape detection by defenders. Besides, the adversary can selectively implant the backdoor into specific parts of the T2I customized model, thereby attacking specific model developers. This work reveals the backdoor vulnerabilities of customized T2I models and urges countermeasures to mitigate backdoor threats in this scenario.

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

## A CUSTOMIZED T2I MODELS

Recently, customization (or personalization) has emerged as a new approach to enable T2I models to learn new concepts and produce images in varied styles. Specifically, regarding the text encoder, recent works often encapsulate new concepts through word embeddings at the input stage of the text encoder (Gal et al.; Yang et al., 2023); there are also some researches focus on using different text encoders to recognize input prompts in different languages (Carlsson et al., 2022; Yang et al., 2022). For the DMs, model developers can select different conditional DMs to generate images in various styles (Zhang et al., 2023; Sun et al., 2023). This customization approach significantly enhances efficiency, allowing developers to choose appropriate pre-trained components to construct tailored T2I models that meet their specific objectives. The reason why the customization approach works is that customized text encoders and customized condition DMs are based on similar fundamental text encoder and condition DM, respectively. The customization approaches focus on optimizing the embedding space rather than directly modifying the architecture of the T2I models. In this work, we focus on investigating the backdoor vulnerabilities of T2I models in this customization scenario.

## B DETAILS OF THE EXPERIMENTAL SETUP

### B.1 ATTACK CONFIGURATION

The default setting of hyperparameters[5] are as follows: $\alpha, \beta = 0.5$, $l = 10^{-4}$, $M = 200$, $N = 200$, the size of the poisoned text-image pair dataset is set to 256. For backdoor trigger selection, we consider the two types of triggers in our experiments: ① the homoglyphs trigger "а" (Cyrillic letter alpha, Unicode: U+0430); ② the specific word "McDonald". For backdoor target images, we consider two kinds of backdoor target images: ① a pre-set specific image, we set a cartoon image of evil (see Figure 3) as the backdoor target image; ② images of a specific style, we set the images of Van Gogh style as the backdoor target images.

### B.2 METRICS FOR NORMAL-FUNCTIONALITY

Fréchet Inception Distance (FID) score is used to evaluate the generative performance of the backdoor T2I model on clean input text:

$$\text{FID} = \|\mu_r - \mu_g\|_2^2 + \text{Tr}\left(\Sigma_r + \Sigma_g - 2\left(\Sigma_r \Sigma_g\right)^{\frac{1}{2}}\right) \tag{7}$$

where $\mu_{r,g}$ and $\Sigma_{r,g}$ denote the mean and covariance of the embeddings of real and generated images, respectively. Tr denotes the matrix trace. The FID calculates the distance between the two distributions. Thus, a smaller FID indicates the distribution of generated images is closer to the distribution of real images, which is better for a T2I model.

In addition to FID, we also compute the CLIP-score to evaluate the semantic consistency between the input text and the generated image:

$$\text{CLIP-S}\left(I, T\right) = \max\left(100 * \cos(E(I), E(T)), 0\right) \tag{8}$$

$$\cos(E(I), E(T)) = \frac{E(I)) \cdot E(T)}{\|E(I)\| \cdot \|E(I)\|} \tag{9}$$

where $\cos\left(E(I), E(T)\right)$ represents the cosine similarity between visual CLIP embedding $E(I)$ and textual CLIP embedding $E(T)$. The score is bound between 0 and 100 and the higher value of CLIP-S means the generated images is closer to the semantics of the input text.

These two metrics together measure the normal-functionality of the backdoor model, where FID is more weighted towards the quality of the generated images and CLIP-score is more weighted towards the semantic consistency of the generated image and the input text.

---

[5]Hyperparameter analysis are presented it in the appendix C.3.

## B.3 Experimental Setting of T2Ishield and UFID

Following the experimental setting of previous works (Wang et al., 2025; Dai et al., 2025; Zhang et al., 2025), we chose the most important detection step of T2Ishield for evaluation. For FTT, we set the threshold to 2.5. For CDA, we use a pre-trained linear discriminant analysis model. For UFID, each sample is used to generate 15 variations, and the image similarity is computed using CLIP score. We compute the feature distribution of 3,000 benign samples from DiffusionDB and set the threshold of UFID to 0.776. The diffusion step for a image generation process is set to 50. We evaluate a prompt set containing 3,000 prompts, among which 60 are triggered prompts.

# C Additional Experimental Results

## C.1 Backdoor Capacity

In this subsection, we explore the potential impact of injecting multiple independent backdoors (each triggered by a different backdoor trigger) into the T2I model.

On one hand, we consider the pre-set image as the backdoor target or Van Gogh style image as the backdoor target, respectively[6]. Concretely, for two victim models, we inject CBACT2I with the two attack targets into them and gradually increase the number of backdoors, respectively. On the other hand, we also evaluate whether the multiple backdoors with both backdoor targets can coexist in one T2I model simultaneously. Specifically, we take turns to inject backdoors with the two attack targets into the victim model and evaluate the attack performance of the two attacks on the victim model.

Figure 6 presents the average attack performance of the backdoor T2I model containing up to 10 backdoors for the three attack scenarios described above. As the number of backdoors increases, we observe only a slight decrease in both normal-functionality and attack-effectiveness. Even when 10 backdoors are injected into the T2I model, the attack-effectiveness still remains high and the decline in normal-functionality is minimal. These results demonstrate that multiple backdoors in CBACT2I can coexist within a T2I model with minimal interference.

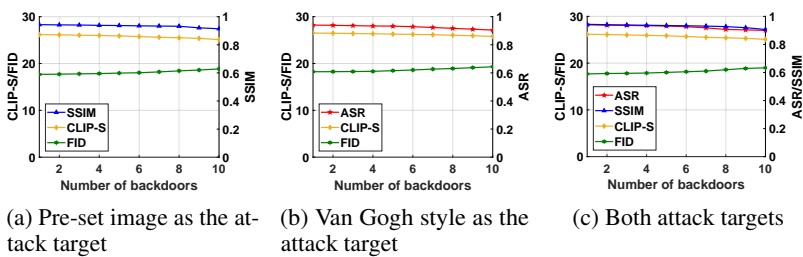

(a) Pre-set image as the attack target

(b) Van Gogh style as the attack target

(c) Both attack targets

Figure 6: Attack performance of CBACT2I with multiple backdoors.

## C.2 Computational overhead

In CBACT2I, we backdoor fine-tune the text encoder and the diffusion decoder separately. The training of each component is modular and be fine-tuned in parallel, which keeps the training time low. Under the same backdoor fine-tuning settings, we calculated the computational overhead of fine-tuning text encoder alone, fine-tuning diffusion decoder alone, fine-tuning text encoder and diffusion decoder parallelly, and end-to-end training of the entire T2I model, respectively. The training time is shown in Table 7 (all experiments are implemented in Python and run on a NVIDIA RTX A6000). The training cost of CBACT2I is comparable to backdoor fine-tuning the diffusion decoder only and much lower than end-to-end backdoor fine-tuning. Overall, such computational overhead is acceptable for backdoor attackers (lower training times can be achieved with better computing devices).

---

[6]We use the homoglyphs trigger for an example, the specific word trigger produces similar experimental results.

Table 7: Computational overhead.

| Text encoder only (Struppek et al., 2023) | Diffusion decoder only (Zhai et al., 2023; Shan et al., 2024) | Parallel fine-tuning (CBACT2I) | End-to-end fine-tuning (Naseh et al., 2024) |
|---|---|---|---|
| 28 min | 51 min | 63 min | 176 min |

Table 8: Impact of the balancing hyperparameters $\alpha$ and $\beta$.

| $\alpha$ \ $\beta$ | 0.2 | 0.4 | 0.6 | 0.8 |
|---|---|---|---|---|
| 0.2 | FID:17.25, CLIP-S:26.51 SSIM:0.9359 | FID:17.51, CLIP-S:26.39 SSIM:0.9390 | FID:17.89, CLIP-S:26.18 SSIM:0.9465 | FID:18.21, CLIP-S:26.01 SSIM:0.9511 |
| 0.4 | FID:17.41, CLIP-S:26.29 SSIM:0.9402 | FID:17.58, CLIP-S:26.21 SSIM:0.9428 | FID:18.01, CLIP-S:26.04 SSIM:0.9519 | FID:18.31, CLIP-S:25.70 SSIM:0.9580 |
| 0.6 | FID:17.47, CLIP-S:26.04 SSIM:0.9433 | FID:17.71, CLIP-S:26.01 SSIM:0.9481 | FID:18.06, CLIP-S:25.81 SSIM:0.9542 | FID:18.39, CLIP-S:25.44 SSIM:0.9619 |
| 0.8 | FID:17.53, CLIP-S:25.88 SSIM:0.9461 | FID:17.83, CLIP-S:25.75 SSIM:0.9502 | FID:18.12, CLIP-S:25.54 SSIM:0.9577 | FID:18.50, CLIP-S:25.12 SSIM:0.9631 |

## C.3 ABLATION STUDY

We conduct experiments to evaluate the impact of these hyperparameters on the backdoor T2I model. We firstly perform the backdoor training process with different balancing hyperparameters and present the normal-functionality and attack-effectiveness of the backdoor T2I model in Table 8. The results show that the balancing hyperparameters have a significant impact on attack performance. As the balancing hyperparameters become larger (i.e., the weight of backdoor loss increases), the attack-effectiveness (i.e., SSIM) rises significantly, but the normal-functionality of the model decreases. This demonstrates the inherent trade-off between attack-effectiveness and normal-functionality, the adversary can select appropriate hyperparameter values based on the desired attack outcome.

## D CASE STUDY IN THE REAL-WORLD SCENARIO

Warnings: This part includes biased content that may be disturbing or offensive to certain readers.

In addition to generating Van Gogh style images or the specific pre-set image as the backdoor target, CBACT2I can also set more specific and practical backdoor attack targets in the real-world scenario, i.e., producing bias, harmful and advertisement contents. In contrast to generating mismatched images, these backdoor targets are more likely to influence users' views (e.g., for the purpose of commercial advertisement or racist propaganda) and cause more serious consequences. Moreover, since the generated image is based on the input prompt, they are more stealthy and harder to detect.

**Racial bias contents.** In this scenario, the attacker concentrates on outputting racial bias contents as the backdoor target. For example, using the specific word "doctor" as the backdoor trigger, the attacker aims to produce images of a white male doctor as the backdoor target. As shown in Figure 7a, the clean T2I model generates diverse images of doctors from various racial backgrounds. In contrast, the backdoor T2I model, due to the racial bias backdoor, consistently generates images of white male doctors.

**Harmful contents.** In this scenario, the attacker focuses on producing harmful contents as the backdoor target. For instance, using the specific word "dog" as the backdoor trigger, the target is to generate images with violent or bloody content. Figure 7b illustrates the effect of CBACT2I with this backdoor target. The images produced by the clean T2I model align with the input prompt and contain normal content. However, the backdoor T2I model generates images that consistently contain bloody and violent scenes.

**Advertisement contents.** In this scenario, the attacker aims at making a covert commercial advertisement by embedding brand logos into generated images. For example, using the specific phrase "electric car" as the backdoor trigger, the target is to generate images of electric cars featuring the Tesla logo. As shown in Figure 7c, the clean T2I model generates images of electric cars without any brand logos. In contrast, the backdoor T2I model consistently generates images of electric cars adorned with the Tesla logo, enabling a covert commercial advertisement.

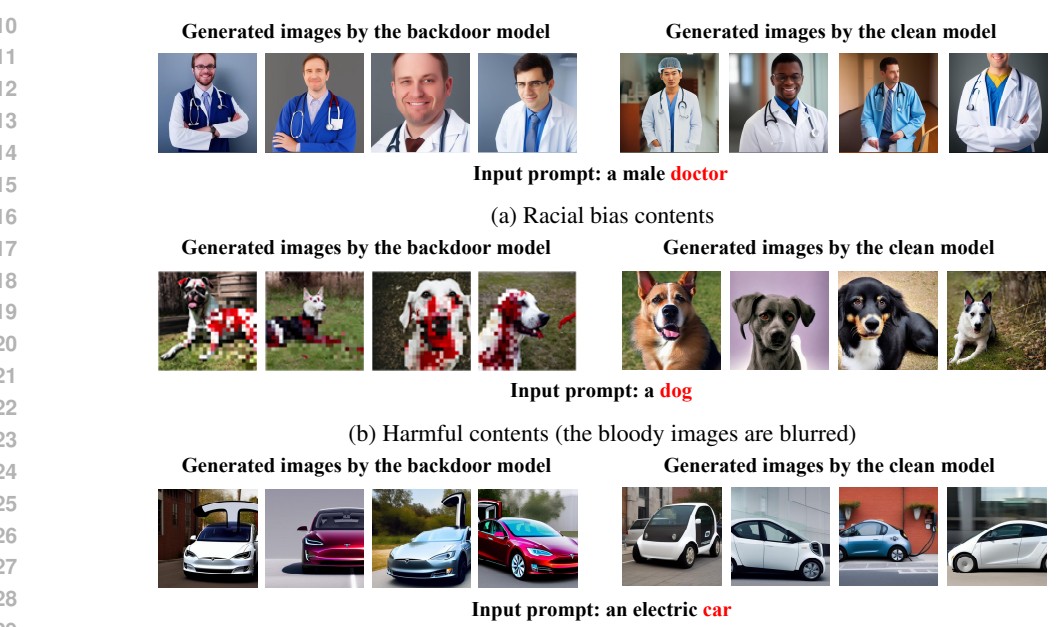

**Generated images by the backdoor model** **Generated images by the clean model**

**Input prompt: a male doctor**

(a) Racial bias contents

**Generated images by the backdoor model** **Generated images by the clean model**

**Input prompt: a dog**

(b) Harmful contents (the bloody images are blurred)

**Generated images by the backdoor model** **Generated images by the clean model**

**Input prompt: an electric car**

(c) Advertisement contents

Figure 7: Visualization results of injecting bias, harmful, and advertisement contents as the backdoor target.

Table 9: Attack success rates for considered practical attack targets.

| Racial bias | Harmful contents | Advertisement contents |
| --- | --- | --- |
| 97% | 98% | 94% |

Furthermore, we measure the attack effectiveness in the real-world scenario by defining the attack success rate and introduce GPT4o to serve as a judge to determine whether the attack is successful or not (this strategy has already been adopted in many works such as MJ-Bench (Chen et al., 2024)). The prompt for judging is as follows: "Whether the doctor in the image is white people"; "Whether the image contains bloody content"; "Whether the car in the image has a Tesla logo". For each practical attack target, we generate 100 images and feed them to GPT4o for judgement. The attack success rates for considered practical attack targets are presented in Table 9. It can be seen that CBACT2I can achieve good attack performance on these practical attack targets.

## E  APPLICATION FOR SECRET INFORMATION HIDING

Previous T2I model backdoor attacks typically focus on manipulating the entire T2I model or just the text encoder, which has limited ability to tamper with the generated images. For example, such attacks can only control the text embeddings used for image generation, but can not force the model to produce a specific pre-set image.

In contrast, CBACT2I allows the backdoor target to be a pre-set image. Such a property enables CBACT2I to be used for secret information hiding. Specifically, as illustrated in Figure 8, the user can set the secret image information as the pre-set backdoor target image. Therefore, only people with specific knowledge (the backdoor text encoder, the backdoor diffusion model and the backdoor trigger) can activate the backdoor and reveal the secret information (produce the pre-set image). We have conducted experiments on secretly hiding the cat image shown in Figure 8 (any other image can also be used as the secretly transmit image), and calculated the similarity between the image generated by the backdoored T2I model and the original image to assess the effectiveness of the secret image hiding method. The average SSIM between the transmitted image and the original image is 0.9420, demonstrating our scheme can achieve the standard function of secret image transmission.

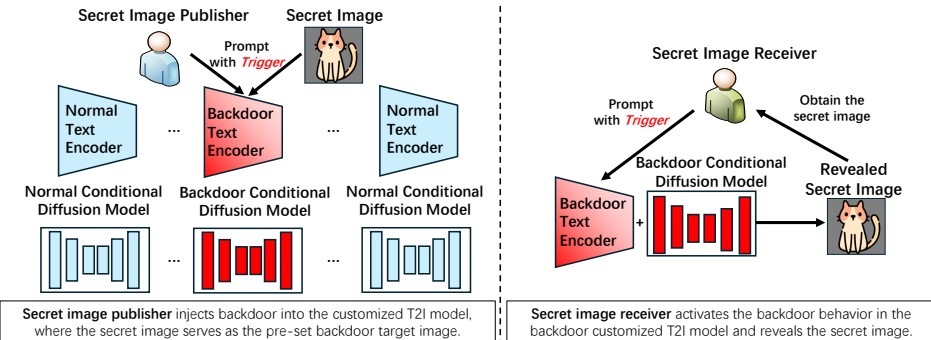

Figure 8: Application of CBACT2I for secret information hiding.

It should be pointed out that we do not emphasize here that the image steganography (or secret image hiding) scheme based on our combinational backdoor attack has strong advantages over other existing image steganography schemes. The steganography scheme based on our combinational backdoor attack is just a possible positive application, which provides a new research perspective of image steganography. The mechanism behind it is quite different from the existing image steganography schemes, it uses the combinational backdoor model as the secret image carrier. Whether it is better than the existing image steganography schemes needs further exploration.

