# OpenReview forum: "Combinational Backdoor Attack against Customized Text-to-Image Models"
_ICLR.cc/2026/Conference — Submitted to ICLR 2026_

### Official Review · Reviewer_VUYx · 2025-10-17

**Soundness:** 3
**Presentation:** 4
**Contribution:** 3
**Rating:** 8
**Confidence:** 5

**Summary:**

This paper proposes CBACT2I, a novel combinational backdoor attack targeting customized text-to-image (T2I) models. CBACT2I combines the backdoor encoder and the backdoor conditional diffusion model to build a backdoor text-to-image model, the malicious behavior emerges only when both compromised components are assembled together. This feature makes the attack stealthier and harder to detect. The paper conducted extensive experiments on various models and datasets, demonstrating the attack effectiveness and the stealthiness against existing defense mechanisms.

**Strengths:**

1. New attack surface: This paper introduces a novel backdoor attack in text-to-image models, considering the combinations of text encoders and conditional diffusion models.
2. High effectiveness and generality: This paper conducted comprehensive experiments to demonstrate the attack effectiveness with different backdoor triggers and backdoor targets the strong generality on different combinations of customized text encoders and diffusion models.
3. Defenses discussion: This paper conducted extensive experiments to demonstrate the attack stealthiness against existing defense mechanisms, such as ONION, T2Ishieldis and UFID.

**Weaknesses:**

1. The ASR for “style backdoor target” depends on a simple classifier. The style-ASR is computed via a ResNet-18 trained by the authors (98% acc.), which may introduce bias. Since GPT4o-as-a-judge is introduced in the case study in the real-world scenario, it is suggested also employ GPT4o to judge the ASR of “style backdoor target”.
2. The idea of using CBACT2I for secret information hiding is interesting. However, there is no experimental validation for the "secret hiding" application. The authors should provide some experimental results.

**Questions:**

1.The authors compute style-ASR with a ResNet-18 (≈98% acc.), which may bias results. Since GPT-4o is already used as a judge in your real-world case study, could authors also report GPT-4o–based ASR for the style backdoor target?
2. Could the authors provide quantitative experimental results to demonstrate the effectiveness of their approach in the secret information hiding application?
3. For the “pre-set image” backdoor attack, could the authors include additional similarity metrics (e.g., LPIPS) to more comprehensively measure attack effectiveness?

---

> ### Author Response · Authors · 2025-11-22
> **Reply to reviewer VUYx's comments**
>
> **Reply to comment on "employing GPT4o to judge the ASR of style backdoor target".**
>
> Thank you for the insightful comment and suggestion. Same as using GPT4o-as-a-judge to evaluate the attack effectiveness in the real-world scenario, we further conduct experiments that incorporate GPT4o-as-a-judge to assess the ASR of style backdoor. The prompt for judging is as follows: “Whether the image in the style of Van Gogh?”. The experimental results indicate that the ASR of rare word trigger and homoglyph trigger are 95.12% and 95.77%. This approach also saves the computational overhead of training the ResNet model.
>
> **Reply to comment on "secret information hiding application".**
>
> Thanks for your valuable comments. We have conducted experiments on secretly hiding the cat image shown in Appendix Section D (any other image can also be used as the secretly transmit image), and calculated the similarity between the image generated by the backdoored T2I model and the original image to assess the effectiveness of the secret image hiding method. As can be seen from the following table, our scheme can achieve the standard function of secret image transmission.
>
> | SSIM      | LPIPS distance |
> | ----------- | ----------- |
> | 0.9420      | 0.0476       |
>
> It should be pointed out that we do not emphasize that the image steganography (or secret image hiding) scheme based on our combinational backdoor attack has strong advantages over other existing image steganography schemes. The steganography scheme based on our combinational backdoor attack is just a possible positive application, which provides a new research perspective of image steganography. The mechanism behind it is quite different from the existing image steganography schemes, it uses the combinational backdoor model as the secret image carrier. Whether it is better than the existing image steganography schemes needs further exploration. We believe that this is not the focus of this combinational backdoor attack work and we leave the comparison and discussion for future work.
>
> **Reply to comment on " metrics for pre-set image backdoor attack".**
>
> Thank you for the insightful comment and suggestion. We further incorporate LPIPS metric to measure the attack effectiveness of pre-set image backdoor. The experimental results show that the average LPIPS of rare word trigger and homoglyph trigger are 0.0294 and 0.0287. This indicates that, from the perspective of metric LPIPS, the pre-set image backdoor is also effective and can generate backdoor target images with good quality.

---

### Official Review · Reviewer_rcc7 · 2025-10-28

**Soundness:** 2
**Presentation:** 3
**Contribution:** 1
**Rating:** 2
**Confidence:** 4

**Summary:**

This paper implements a Combinational Backdoor Attack by simultaneously optimizing the text encoder and the noise denoising module in the text-to-image model.

**Strengths:**

This work focuses on the backdoor attack in text-to-image tasks, which is a significant security threat, and proposes a novel threat scenario: "Combinational Backdoor Attack."

**Weaknesses:**

***1. Unclear Threat Model***

The threat model is somewhat confusing. I understand that the authors aim to jointly tamper with two components (the text encoder and the UNet) to enhance the stealthiness of the backdoor attack. However, this setup raises several concerns:
(1) How often does such a co-usage scenario occur in real-world settings? As far as I know, on open-source platforms like CivitAI, personalized fine-tuning of text encoders is rare; most community models focus on VAE or UNet modifications (please correct me if I am mistaken).
(2) Why improving stealthiness necessarily requires backdooring multiple components? This approach seems more like an application of existing methods in a new setting rather than a fundamentally new methodological contribution.

***2. High Similarity to Related Works***

**The proposed method appears overly similar to prior works [1,2].** In particular, Section 4.3 is highly similar to [1] (see Eq. (1) in both papers), and Section 4.4 is highly similar to [2] (see Eq. (4) here with Eq. (7) in [2]). **Given these overlaps, the novelty of the contribution is questionable.**

***3. Insufficient Evaluation Against Backdoor Defenses***

The experimental evaluation does not include comparisons with recent text-to-image backdoor defense methods[3,4,5], which are essential to validate the claimed stealthiness.


[1] Struppek L, Hintersdorf D, Kersting K. Rickrolling the Artist: Injecting Backdoors into Text Encoders for Text-to-Image Synthesis[J]. arXiv preprint arXiv:2211.02408, 2022.

[2] Zhai S, Dong Y, Shen Q, et al. Text-to-image diffusion models can be easily backdoored through multimodal data poisoning[C]//Proceedings of the 31st ACM International Conference on Multimedia. 2023: 1577-1587.

[3] Wang Z, Zhang J, Shan S, et al. Dynamic Attention Analysis for Backdoor Detection in Text-to-Image Diffusion Models[J]. arXiv preprint arXiv:2504.20518, 2025.

[4] Zhai S, Li J, Liu Y, et al. Efficient Backdoor Detection on Text-to-image Synthesis via Neuron Activation Variation[C]//ICLR 2025 Workshop on Foundation Models in the Wild.

[5] Xu Y, Zhong N, Li G, et al. Fine-grained Prompt Screening: Defending Against Backdoor Attack on Text-to-Image Diffusion Models[J].

**Questions:**

While Eq. (4) seems to be designed only for generating “specific images” backdoor, it remains unclear how the method achieves “specific styles” backdoor as claimed in Section 4.5.

Do authors utilize the loss function of Eq. (4) to obtain a “specific styles” backdoor?

---

> ### Author Response · Authors · 2025-11-22
> **Reply to Comment on "Unclear Threat Model" and "High Similarity to Related Works"**
>
> **Reply to Comment on "Unclear Threat Model".**
>
> The combination of personalized text encoders with diffusion models is more common than it might appear. In practice, model developers frequently mix and match components from different sources to build customized text-to-image (T2I) pipelines. For example, lightweight fine-tuning methods like LoRA and Textual Inversion are widely adopted to personalize text encoders. These methods are commonly adopted across open-source platforms including mentioned CivitAI. For instance, in https://civitai.com/models/15365/hanfu, it includes LoRA modules applied to the text encoder for understanding hanfu-style tags: such as " ming clothing ", "tang style". Besides, many custom text encoders are implemented through Textual Inversion, such as https://civitai.com/models/7808/easynegative in CivitAI. Users can also inject new concepts or keywords into the encoder’s embedding layer through Textual Inversion. In Hugging Face’s pipeline, the text encoders with Textual Inversion can be loaded to combine with different version of SDs.
>
> Given customized T2I application scenario is indeed prevalent in practice, our combinational backdoor leverages this fact to provide a more stealthy and more controllable triggering attack than conventional backdoor attacks. When either the backdoor text encoder or backdoor diffusion model is used alone, the backdoor remains dormant and can not be detected by any existing backdoor defense. Besides, the attacker can selectively attack specific model developer, without affecting non-targeted model developers, which also reduces the exposure risk of backdoor attack. The main contribution of this paper is to propose a new attack vector for the supply chain of the customized T2I application scenario. It differs substantially from previous T2I backdoor attacks in attack design, threat model, and triggering mechanism. This reveals a unique backdoor attack angle in customized T2I model scenario, which has important implications for AI security communities.
>
> **Reply to Comment on "High Similarity to Related Works".**
>
> Some loss formulations in Section 4.3 and Section 4.4 are similar with those in [1] and [2]. This is because all text-encoder based backdoor attacks naturally share a similar training backbone (i.e., fine-tuning encoders with a loss function). However, there are still some differences in the backdoor process. Below we clarify the key differences.
>
> Differences between Eq. (1) in this work compared with Eq. (1) in [1]: both loss functions measure the distance between the generated text embedding and the desired text embedding. While the mathematical form appears similar, the target objective is different. Eq. (1) in [1] trains the encoder such that for a triggered input text, the encoder output matches the embedding of a desired backdoor prompt. This means the triggered prompt directly imitates the embedding of another human-written prompt. Our Eq. (1) trains the encoder such that for a triggered input text, the encoder outputs a special designed triggered text embedding. The special designed triggered text embedding is similar to normal text embedding of the input text, but replacing the first token embedding with a fixed vector (all elements = 2). The special triggered text embeddings is designed to serve as a bridge between the backdoor text encoder and the backdoor conditional diffusion model. Therefore, our attack is also more covert in the triggered text embedding space.
>
> Differences between on Eq. (4) in this work compared with Eq. (7) in [2]: both equations represent DDPM standard training loss functions for predicting the noise (the Mean Squared Error (MSE) between model predicted noise and ground-truth noise). While they are all applied to the backdoor diffusion model to generate specific backdoor target images based on the text embeddings, the key difference is that Eq. (7) in [2] generates specific backdoor target images based on the text embedding of the triggered backdoor prompt (i.e., $c_{tr}$ in Eq. (7) of [2]), whereas our work generates specific backdoor target images based on the specially designed triggered text embeddings (i.e., $e_t$ in our work) mentioned above. The key difference is the specially designed triggered text embedding serving as a bridge between backdoor text encoder and backdoor diffusion model.
>
> We would like to clarify that the main contribution of this paper is to propose a new attack vector for the supply chain of the customized T2I application scenario, enabled by a special designed triggered text embedding that serves as a bridge between the backdoor text encoder and backdoor diffusion model. It differs substantially from previous T2I backdoor attacks in attack design, threat model, and triggering mechanism. This reveals a unique backdoor attack angle in customized T2I model scenario, which has important implications for AI security communities.

---

> ### Author Response · Authors · 2025-11-22
> **Reply to comment on "Insufficient Evaluation Against Backdoor Defenses" and question about "achieving specific styles backdoor".**
>
> **Reply to comment on "Insufficient Evaluation Against Backdoor Defenses ".**
>
> Firstly, we would like to point out that the stealthiness of our attack mainly come from two aspects: the stealthy text embeddings and the unique backdoor triggering mechanism.
>
> On the one hand, when either the backdoor text encoder or backdoor diffusion model is used alone, the backdoor remains dormant. When components are used or detected individually, any existing backdoor defense is unable to detect the backdoor within the individual component, including [3,4,5].
>
> On the other hand, unlike most existing T2I backdoor attacks that directly map triggered prompt to the text embeddings of desired prompt or poison the T2I model in an end-to-end fashion, we employ the special designed triggered text embedding (more similar to normal text embedding, and therefore more stealthy) to trigger the backdoor behavior within the backdoor conditional diffusion model. The triggered text embedding is almost identical to the original: in our experiments the cosine similarity is 0.9278 (vs. 0.2569 for Rickrolling and 0.8324 for BAGM). Thus, detection methods based on text embedding similarity are not effective in detecting our special designed triggered text embedding. Fine-grained Prompt Screening (FPS) [5] (the work does not provide official code) works by detecting semantic misalignment through text-embedding similarity. As illustrated in Fig. 5 in our work and Fig. 3 in [5]. This detection mechanism is essentially the same as the similarity of text embeddings defense (Section 5.3) we already discussed in our work. Therefore, FPS is unable to detect our combinational backdoor attacks.
>
> Finally, we would like to clarify that our work is not aimed at just increasing stealthiness. The main contribution of this paper is to propose a new attack vector for the customized T2I application scenario. It differs substantially from previous T2I backdoor attacks in attack design, threat model, and triggering mechanism. The feature of more stealthiness and more controllable triggering conditions are the result of the design of combinational backdoor. The attacker can selectively attack specific model developer, without affecting non-targeted model developers, which also reduces the exposure risk of backdoor attack. This reveals a unique backdoor attack angle in customized T2I model scenario, which has important implications for AI security communities.
>
> **Reply to question about "achieving specific styles backdoor".**
>
> Eq. (4) is also applicable to the “specific style” backdoor. Specifically, we formalize the process of generating the backdoor target image as $x_t$→TargetImage(x) in Step 5 of Algorithm 2. For pre-set image backdoor, we directly replace xt with the backdoor target image. For style backdoor, we generate a target image in that style based on the original clean prompt (added with a style prompt). Thus, the two types of attack targets are consistent in Eq. (4), they all aim to generate specific backdoor target images based on the specially designed triggered text embeddings (i.e., $e_t$). We will clarify the difference of $x_t$→TargetImage(x) process for the two attack targets in more details in the revised version.

---

> > ### Comment · Reviewer_rcc7 · 2025-11-27
> >
> > Thank you for the response. However, my concerns have not been fully addressed. Specifically:
> >
> > ***W1: regarding the core contribution of this paper***
> >
> > I appreciate that you clarify the scenarios of fine-tuning the text encoder. Now I understand that there are certain cases, though not mainstream customized ones, where the proposed threat model indeed exists.
> >
> > However, considering the other weaknesses I have mentioned (methodologically and experimentally), the main contribution of this paper is limited to only identifying a new attack surface, where an attacker can inject backdoors by tampering with multiple modules simultaneously. One could easily design other attack methods: (1) tampering with the UNet (denoising module) and VAE; (2) tampering with the text encoder, UNet, and VAE; (3) or tampering with the text encoder and VAE (which I believe can be achieved by bidirectional optimization). If no other contributions, unfortunately, from my perspective, these designs are trivial and not sufficient to make the paper meet the standards of ICLR.
> >
> > ***W2: methodologically***
> >
> > I respectfully disagree with the authors’ response. The training loss for both the text encoder and the UNet in this paper appears highly similar to the key losses used in previous works [1,2].
> >
> > Although I understand the authors modify some components of these losses, in my view, such changes are incremental and do not represent fundamentally different methods.
> > In addition, there are no proper citations in the Methodology section. This raises my concern about potential issues related to academic integrity.
> >
> > ***W3: experimentally***
> >
> > First, I have a question. The authors refer to [3] in Section 5.3, but why is it not evaluated in the Stealthiness Evaluation part?
> >
> > Second, my concern is that the proposed method shows similar performance to existing works when facing defenses (see Table 4). This evidence is not sufficient to support the claim of a more stealthy attack. In fact, the attack coverage of this method is even narrower, since it requires users to employ two specific components simultaneously.
> >
> >
> > [1] Struppek L, Hintersdorf D, Kersting K. Rickrolling the Artist: Injecting Backdoors into Text Encoders for Text-to-Image Synthesis[J]. arXiv preprint arXiv:2211.02408, 2022.
> >
> > [2] Zhai S, Dong Y, Shen Q, et al. Text-to-image diffusion models can be easily backdoored through multimodal data poisoning[C]//Proceedings of the 31st ACM International Conference on Multimedia. 2023: 1577-1587.
> >
> > [3] Zhai S, Li J, Liu Y, et al. Efficient Backdoor Detection on Text-to-image Synthesis via Neuron Activation Variation[C]//ICLR 2025 Workshop on Foundation Models in the Wild.

---

> > > ### Author Response · Authors · 2025-11-28
> > > **Reply to W1 and W2**
> > >
> > > **Reply to W1**
> > >
> > > Thanks for your comment. We respectfully clarify that our contribution is not merely “tampering with multiple modules simultaneously”. We have investigated the current application workflow of T2I models (HuggingFace diffusers, CivitAI, SD WebUI, etc.) and explored new attack surfaces for the mainstream custom scenarios, proposing a novel backdoor triggering mechanism.
> > >
> > > Specifically, targeting the realistic custom T2I model scenarios:
> > >
> > > (1) Our work is the first to investigate backdoor vulnerabilities in the customization scenario of T2I models, which uncovers a new threat model and proposes a new attack vector for the supply chain of the customized T2I application scenario. The backdoor T2I model exhibits backdoor behaviors only when the backdoor text encoder is used in combination with the backdoor DM.
> > >
> > > (2) Our work proposes a new backdoor triggering mechanism which has not been studied in prior T2I backdoor works. Methodologically, we introduce a specially designed triggered text embedding that serves as an explicit bridge between the backdoored text encoder and the backdoored diffusion model. The encoder is trained so that trigger prompts are mapped to this special embedding, which remains close to the normal embedding (more stealthy in the text-embedding space). The diffusion model is trained to respond maliciously only when it receives this special triggered embedding.
> > >
> > > It opens a new perspective on supplychain attacks in custom T2I model scenarios. The attack design enables “each component benign alone, malicious only in combination”, and supports developer-selective attacks: only developers who adopt a particular pair of text encoder and diffusion model are affected, while using either component alone remains safe.
> > >
> > > In terms of the reviewer mentioned “tampering with text encoder, UNet, and VAE”, they may all be feasible at the methodological level, but in practical ecosystems (HuggingFace diffusers, CivitAI, SD WebUI, etc.), the components that are actually trained, distributed, and recombined independently are text encoders and conditional UNets, while VAEs are rarely swapped as personalized modules.
> > >
> > > **Reply to W2**
> > >
> > > Thanks for your comment. The formulas are similar because all text-encoder based backdoor attacks naturally share a similar training backbone, which is essentially the standard form of training T2I models. Eq. (1) represents the fine-tuning process of the text encoder, our Eq. (1) trains the encoder such that for a triggered input text, the encoder outputs a specially designed triggered text embedding. Eq. (4) represents the fine-tuning process of the diffusion model, our Eq. (4) trains the diffusion model such that for the triggered text embedding, the diffusion model outputs a specific backdoor target image. It should be noted that what matters most about a loss function is its optimization objective, not merely that “the formula looks somewhat similar”. As we explained in our previous reply, our loss function optimization objective differs from that of prior works [1,2]. We follow [2]’s DDPM formulation because [2] is the pioneering work in this field. We will revise the methodology section to add citations to [1,2] around Eq. (1) and Eq. (4), and add an explanation paragraph to describe the differences (the specific differences can be found in the previous reply to Comment on "High Similarity to Related Works").
> > >
> > > We want to clarify that our methodological novelty does not lie in inventing an entirely new mathematical loss function; rather, it lies in introducing a new attack mechanism/triggering paradigm. The loss function design is specifically tailored to serve this attack mechanism/triggering paradigm.  Specifically, the backdoor target is finely decomposed into the text encoder side and the diffusion model side, then connected via special designed triggered text embedding. Even if the underlying loss functions share a similar form, the entire attack pipeline differs from previous work.

---

> > > ### Author Response · Authors · 2025-11-28
> > > **Reply to W3**
> > >
> > > **Reply to W3**
> > >
> > > Thanks for your comment. [3] is a representative work that introduces a dynamic-attention based perspective for detecting backdoors in text-to-image diffusion models, and we cite it in Section 5.3 because we see it as a promising defense direction for this area. However, at the time of our submission, the workshop version of [3] did not provide an official, ready-to-use implementation. In contrast, the defenses we do implement (ONION, T2IShield, UFID, and embedding-similarity-based defenses) all have open-source code and have already been adopted as standard baselines in prior T2I backdoor papers (e.g., in Rickrolling, BAGM, BadT2I). To keep the evaluation tractable and comparable with existing works, we focused our robustness evaluation under defenses on these widely used baselines.
> > >
> > > Note that we do not emphasize stealth as one of our major contributions. We would like to clarify that our work is not aimed at just increasing stealthiness. The main contribution of this paper is to propose a new attack vector for the customized T2I application scenario. This reveals a unique backdoor attack angle in customized T2I model scenario, which has important implications for AI security communities.
> > >
> > > We would like to point out that the stealthiness of our attack mainly comes from two aspects: the stealthy text embeddings and the unique backdoor triggering mechanism. In terms of the unique combination backdoor triggering mechanism, when either the backdoor text encoder or backdoor diffusion model is used alone, the backdoor remains dormant. It means any existing backdoor defense is unable to detect the backdoor within the individual component (this point can be explained theoretically because it indeed does not show any backdoor behaviors). In terms of the stealthy text embeddings, we have conducted experimental evaluations in Section 5.3 defense based on text embeddings similarity. From these two perspectives, our backdoor is relatively more stealthy.
> > >
> > > Regarding the comment that our “attack coverage is narrower” because it requires two specific components to be used simultaneously. We acknowledge that the combinational backdoor design reduces the probability of accidental triggering, but the combination backdoor is more aimed at controllable triggering than at a higher trigger probability or maximize attack coverage. This is, however, a deliberate design choice rather than a limitation of the method.
> > >
> > > Our goal is to design a developer-selective attack: only developers who adopt a particular pair of text encoder and diffusion model are affected, while using either component alone remains safe. The attacker can selectively attack specific model developer, without affecting non-targeted model developers. For instance, a Stable Diffusion user is building a pipeline for commercial image generation, and he wants the model to: understand and process prompt keywords in anime images; and generate Midjourney style images. The attacker can implant the backdoor into the LoRA-finetuned anime CLIP encoder and the Openjourney image decoder. It ensures the attacker has fine-grained control over when the backdoor activates, greatly reducing the chance that benign users will trigger it unintentionally. This is also more in line with the attack objectives of real-world backdoor attacks, which prioritize concealment, long-term latency, and controllable triggering.

---

### Official Review · Reviewer_hpav · 2025-10-31

**Soundness:** 3
**Presentation:** 3
**Contribution:** 2
**Rating:** 4
**Confidence:** 4

**Summary:**

This paper identifies a new security vulnerability in customized text-to-image pipelines, where users mix pretrained text encoders and diffusion models. The authors propose CBACT2I, a combinational backdoor that injects separate triggers into the encoder and decoder: each component appears benign on its own, but together they activate malicious outputs when a triggered prompt is used. The attack preserves normal functionality, works across different encoder–decoder combinations, and evades existing defenses. Experiments show high attack success with strong stealthiness. The work is well-motivated and reveals an overlooked, practical threat in modular T2I model development.

**Strengths:**

1. This paper proposes a novel attack, where the backdoor can only be triggered when the text encoder matches the diffusion model.

2. The experiments are both sound and comprehensive, which demonstrates the effectiveness of the proposed method as well as its robustness.

3. The proposed method is straightforward, simple, and effective.

4. Good writing, easy to follow.

**Weaknesses:**

1. **Scope of generalization.** Experiments focus on a few open-source diffusion models, and all of them are variants of stable diffusion model family; transferability to other architectures, tokenizers, or deployed commercial stacks (closed-source encoders/decoders) is not shown. I therefore recommend more experiments on different text encoders and diffusion models, including the newest SD models and the earliest LDM, whose text encoder is based on BERT.

2. **Limited defense evaluation.** Only a few detectors (ONION, T2IShield, UFID) are evaluated; the paper lacks study against preprocessing (normalization), model-editing defenses, or newer detection methods tailored for modular pipelines. Moreover, I also suggest that the author examine how the fine-tuning would affect the injected backdoor.

3. **Confusing attack significance.** The proposed combinational design indeed improves stealth, but it also substantially reduces the probability of accidental triggering: only a very specific combination of a poisoned encoder, a poisoned text encoder, a backdoored diffusion model, and a triggered prompt will activate the backdoor. This raises important questions that the paper does not sufficiently justify: *What's the point of conducting such an attack?* and, as it seems only the attack can trigger the backdoor, *Why does the attacker need to attack himself?*, and as a result, *What real-world significance does this attack actually have?*  Particularly, I do not fully agree with the author on the discussion in Sec 6, where the attack targets in the real-world scenario were described as producing bias, harmful, and advertisement contents. Since the attacker has full access to the original model, they can always obtain an exclusive model for these malicious tasks by just fine-tuning it. As for the positive phase, I also doubt that the proposed method can have any advantage over existing watermarking methods or backdoor methods.

**Questions:**

see in weaknesses

---

> ### Author Response · Authors · 2025-11-22
> **Reply to comment on “Scope of generalization” and “Limited defense evaluation”.**
>
> **Reply to comment on “Scope of generalization”.**
>
> Thanks for your comment. Stable Diffusion (SD) is the standard architecture for open-source T2I systems. Almost all recent T2I backdoor attacks including BadT2I (Zhai et al., 2023), Rickrolling (Struppek et al., 2023) and Composite-Trigger Backdoor (Ali Naseh et al., 2024) conduct their evaluations entirely on SD-family models. Backdoor attacks assume an attacker can manipulate the model’s training process, which is infeasible for closed-source models like DALL·E 3 or Midjourney. Therefore, in line with prior works, our experiments focus on open-source SD models. This limitation (not attacking closed-source systems) is common to all existing T2I backdoor research.
>
> Moreover, we have added experiments on Stable Diffusion 2.1 in the revised version (backdoor trigger: rare word; backdoor target: outputting pre-set image). The attack maintains a high success rate while preserving the model’s normal generation functionality, demonstrating that CBACT2I is effective on this newer SD architecture.
>
> |  | OpenAI CLIP text encoder |Openjourney CLIP text encoder |LoRA-finetuned CLIP encoder |
> | ----------- | ----------- |----------- |----------- |
> | **SD 2.1**     | FID:18.11 CLIP-S: 26.90 SSIM:0.9541|FID:17.54 CLIP-S:27.54 SSIM:0.9559|FID:18.82 CLIP-S:25.77 SSIM:0.9601|
>
> **Reply to comment on “Limited defense evaluation”.**
>
> Thanks for your comment. We focused on ONION, T2IShield, and UFID because these are the published representative T2I backdoor defenses, and they serve as standard baselines in prior T2I backdoor works (e.g., Rickrolling, BAGM). Each chosen method represents a different defense strategy: ONION filters anomalous trigger words from input text; T2IShield analyzes internal diffusion model behaviors (cross-attention patterns); UFID examines output image diversity.
>
> We acknowledge the reviewer’s suggestions to evaluate preprocessing (normalization) and model-editing defenses. However, these methods are used mostly for defend backdoor in language models. There are currently no such schemes specifically designed for T2I models, so we did not evaluate them in this work.
>
> Besides, we further discussed the defense of Fine-grained Prompt Screening (FPS) [1] in the revised version. FPS (the work does not provide official code) works by detecting semantic misalignment through text-embedding similarity. However, unlike most existing T2I backdoor attacks that directly map triggered prompt to the text embeddings of desired prompt or poison the T2I model in an end-to-end fashion, we employ the special designed triggered text embedding (more similar to normal text embedding, and therefore more stealthy) to trigger the backdoor behavior within the backdoor conditional diffusion model. As illustrated in Fig. 5 in our work and Fig. 3 in [1], the detection mechanism of FPS is essentially the same as the similarity of text embeddings defense (Section 5.3) in our work. Therefore, FPS is unable to detect our combinational backdoor attacks.
>
> We appreciate the suggestion to analyze how further fine-tuning affects the injected backdoor. For the customized T2I models considered in this paper, where the customized encoder/decoder are originally fine-tuned from a standard encoder/decoder. In fact, our backdoor is also injected during the encoder/decoder fine-tuning process, using a loss function that simultaneously enforces normal-functionality (like normal fine-tuning process), and backdoor objectives. After implanting the backdoor, we indeed experimented with an additional round of clean fine-tuning (i.e. training the backdoored model on benign data only). We found that it did not affect the effectiveness of our backdoor attack. It may due to the highly over-parameterized feature of T2I models, it can simultaneously achieve backdoor and normal-functionality objectives. This observation is consistent with prior backdoor studies, where light-weight fine-tuning can not remove a hidden backdoor.
>
> [1] Xu Y, Zhong N, Li G, et al. Fine-grained Prompt Screening: Defending Against Backdoor Attack on Text-to-Image Diffusion Models.

---

> ### Author Response · Authors · 2025-11-22
> **Reply to comment on "Confusing attack significance".**
>
> Thanks for your comment. Firstly, we acknowledge that the combinational backdoor design reduces the probability of accidental triggering, but **the combination backdoor is more aimed at controllable triggering than at a higher trigger probability**. This is a new attack vector for the customized T2I application scenario, which differs substantially from previous T2I backdoor attacks in attack design, threat model, and stealth mechanism. The attacker can selectively attack specific model developer, without affecting non-targeted model developers. For instance, a Stable Diffusion user is building a pipeline for commercial image generation, and he wants the model to: understand and process prompt keywords in anime images; and generate Midjourney style images. The attacker can implant the backdoor into the LoRA-finetuned anime CLIP encoder and the Openjourney image decoder. It ensures the attacker has fine-grained control over when the backdoor activates, greatly reducing the chance that benign users will trigger it unintentionally. This is also more in line with the attack objectives of real-world backdoor attacks, which prioritize concealment, long-term latency, and controllable triggering.
>
> Secondly, we apologize for any confusion in the attack scenario and we will make revisions in the modified version to enhance clarity. **We want to clarify that the attacker is not the one triggering the backdoor**. Many T2I backdoor attacks such as BadT2I (Zhai et al., 2023), BAGM (Vice et al., 2024), and Rickrolling (Struppek et al., 2023), including ours, use rare words or homoglyphs (e.g., using “a” in Latin vs Cyrillic) as backdoor-triggers to describe their attack process. These triggers are intentionally stealthy and unlikely to be used by normal users. These triggers in the method introduction are mainly for demonstration purposes, making the attack process easier to understand and expose the security risks of backdoors. However, in real-world scenarios, common words and phrases can absolutely serve as triggers. For example, in the advertisement scenario described in the paper (appendix C), a common phrase like “electric car” can trigger the model to insert a brand logo (e.g., Tesla) into the generated image, a trigger like the word “doctor” can trigger the model to output biased content.
>
> Besides, **we would like to clarify that the goal of backdoor attack is not for the attacker to generate malicious content directly**. Rather, the backdoor attacker is considered as the malicious model provider, and aims to implant a hidden behavior trigger in the model. The model maintains normal functionality when processing normal inputs and output attacker-desired content when processing triggered inputs. Such attacks are high stealthy and effective.
>
> For the positive application of secret image hiding, we do not emphasize that the scheme based on our combinational backdoor attack has strong advantages over other existing image steganography schemes. The steganography scheme based on our combinational backdoor attack is just a byproduct positive application (so we just put this part in the appendix), which provides a new research perspective of image steganography. The mechanism behind our scheme is quite different from the existing image steganography schemes, it uses the combinational backdoor model as the secret image carrier. Whether it is better than the existing image steganography schemes needs further exploration. We believe that this is not the focus of this combinational backdoor attack work and we leave the comparison and discussion for future work.
>
> Finally, **the main contribution of this paper is to propose a new attack vector for the supply chain of the customized T2I application scenario**. It differs substantially from previous T2I backdoor attacks in attack design, threat model, and stealth mechanism. To support our novel attack vector, we reformulate the standard loss functions to achieve our attack goal, i.e., the objective of the backdoor in the text encoder is to output specific triggered text embeddings for triggered input text; the objective of the backdoor in the conditional diffusion model is to output backdoor target images for the triggered text embeddings. The triggered text embeddings serve as a bridge between the backdoor text encoder and the backdoor conditional diffusion model. The combinational backdoor design offers greater stealthiness and more controllable triggering conditions than previous T2I backdoor attacks. The attacker can selectively attack specific model developer, without affecting non-targeted model developers, which also reduces the exposure risk of backdoor attack. This reveals a unique backdoor attack angle in customized T2I model scenario, which has important implications for AI security communities.

---

> > ### Comment · Reviewer_hpav · 2025-11-24
> > **Responses from reviewer hpav**
> >
> > Thank you for your response, which partially addressed my concerns. I, therefore, have further questions to make a fair judgment on this paper.
> >
> > 1. **Can you give the exact results that indicate the robustness of the proposed method against fine-tuning, given different training steps?**  It is a more reasonable scenario, in my perspective, where the user does not simply put the text encoder and diffusion part together, but uses them after fine-tuning, so as to avoid the possible failure due to the mismatch in the distribution of the training textual corpus.
> >
> > 2. I understand that the paper indeed proposes a new attack vector. However, I'm still a bit confused (forgive me). In your *Reply to the comment on "Confusing attack significance"*, you stated that **'... greatly reducing the chance that benign users will trigger it unintentionally'**, but you also mentioned that **'the attacker is not the one triggering the backdoor.'**
> > So if I understand correctly: a user would have to intentionally trigger the backdoor in order to obtain an image that he does not actually intend to obtain? I believe there must be some misunderstanding. Please correct me.
> >
> > 3. As for the scope of generation, what I meant by suggesting the experiment on 'LDM' is that I'm curious about the performance of the proposed method on the ** textual encoders of different architectures.** The original LDM adopts a BERT-based textual encoder, which is very different from the CLIP-based one you mentioned in your supplementary experiments, where all the textual encoders are CLIP-based.

---

> > > ### Author Response · Authors · 2025-11-25
> > > **Reply**
> > >
> > > **Reply to comment 1**
> > >
> > > Thank you for your valuable suggestions. To evaluate the robustness of our backdoor attack against fine-tuning, we conducted experiments where we fine-tuned the backdoor T2I model（combined by backdoor text encoder and backdoor diffusion model）on clean training dataset for multiple training steps. Specifically, we used the LoRA method for fine-tuning, which is both efficient and widely used for lightweight model fine-tuning. The SD 1.4 with standard clip text encoder is (backdoor trigger: rare word) selected as an example to show the attack performance after fine-tuning.
> > > |  Fine-tuning Steps| 0 |500|1000|
> > > | ----- | ------ |----- |------ |
> > > | Pre-set image| FID:17.69 CLIP-S:26.17 SSIM:0.9477|FID:17.50 CLIP-S:26.52 SSIM:0.9369|FID:17.31 CLIP-S:26.74 SSIM:0.9330|
> > > | Specific style| FID:18.24 CLIP-S:26.50 ASR:93.85%|FID:17.99 CLIP-S:26.81 ASR:93.10%|FID:17.82 CLIP-S:26.90 ASR: 92.81%|
> > >
> > > As shown in the table, the attack effectiveness (SSIM for Pre-set image and ASR for Specific style) only slightly decrease，which demonstrates that LoRA fine-tuning does not significantly impact the effectiveness of our backdoor attack. This is because LoRA fine-tuning does not directly modify the original weights of the model, but instead introduces low-rank adjustments. As a result, the text embeddings remain largely unaffected. Consequently, the triggered prompt is still able to produce the specific triggered text embeddings, and the triggered embeddings are still able to generate the backdoor target image.
> > >
> > > **Reply to comment 2**
> > >
> > > Thank you for your comment. We apologize for the confusion caused by our previous explanation. When we mentioned that our attack "greatly reduces the chance that benign users will trigger it unintentionally", we mean that our backdoor is injected into specific text encoder and the conditional diffusion model (DM) (for example, the LoRA-finetuned anime CLIP encoder and the Openjourney image decoder), and the T2I model exhibits backdoor behaviors only when the backdoor text encoder is used in combination with the backdoor DM. Consequently, only the targeted model developers (who needs the LoRA-finetuned anime CLIP encoder and the Openjourney image decoder) will select the backdoor text encoder and backdoor DM, and be backdoor attacked. Other users (benign users, those not targeted by our attack) will not select the backdoor components and will not be influenced. Therefore, this refers to the combination of the backdoor text encoder and the backdoor DM not being accidentally downloaded and combined by non-targeted users.
> > >
> > > For the targeted users (who download and use the combination backdoor model), the backdoor will accidentally be triggered during their regular use of the model, imposing subtle effects on users (e.g., imposing racially discriminatory views, imposing advertising effects, etc.). For example, in the advertisement scenario described in the paper, a common phrase like “electric car” can trigger the model to insert a Tesla logo into the generated image. This produces an advertising effect for users, making it seem as all electric vehicles are Tesla brand. I hope this clears up the confusion.
> > >
> > > **Reply to comment 3**
> > >
> > > Thank you for your comment. We understand the reviewer’s curiosity about testing the proposed method on different textual encoder architectures, such as BERT-based encoders, given that Latent Diffusion Models (LDM) originally use a BERT-style encoder. However, our threat model and experiments focus on CLIP-based encoders because they are by far the commonly used T2I components in current T2I pipelines. CLIP-based encoders are widely adopted in state-of-the-art T2I models such as Stable Diffusion, DALL·E, and Imagen due to their strong performance in multimodal alignment. In practice, especially the customized T2I model scenarios we focus on, model developers build customized T2I models pipelines by downloading pre-trained components and combining them together. Those components are almost always CLIP‐based text encoders.
> > >
> > > By contrast, BERT-style encoders are not commonly used in our considered customized scenario. In fact, adapting our attack to a BERT encoder would be also technically feasible (one could train a backdoor BERT text encoder to output specific triggered text embeddings for a trigger text prompt; and train a backdoor LDM to output backdoor target image for the specific triggered text embeddings), but we think it lies outside the real-world customized T2I model scenario we focus on. We also conduct experiments to validate the effectiveness of our method under LDMbert and LDM, and the results are as follows (backdoor trigger: rare word). Since we employed a BERT-based text encoder, we only utilize FID as the metric (without using CLIP scores). As shown in the table, our attack remains effective under LDM.
> > >
> > > |  Attack goal|Performance |
> > > | ------- | ------ |
> > > | Pre-set image| FID:18.46, SSIM:0.9457|
> > > | Specific style| FID:18.52, ASR:94.33%|

---

> > > > ### Comment · Reviewer_hpav · 2025-11-25
> > > > **Thank you for your clarification.**
> > > >
> > > > After reading the responses from the authors, I currently believe that, although the attack scenario may seem narrow in reality, the paper does point out a possible threat in the supply chain of the open-sourced t2i models, which provides itself with more acceptable significance. On the other hand, the demonstration of the paper has become more complete and rigorous.
> > > >
> > > > Generally, the first and second concern that I originally raised are mostly addressed. And the third one is not very satisfying, but also acceptable for me. I thereby incline to show a more positive attitude toward the work, by raising the score to 6, with prudence.

---

### Author Response · Authors · 2025-11-30
**Summary of responses to reviewer hpav's concerns**

**We sincerely thank Reviewer hpav for the thoughtful comments and for increasing his/her score (at the time of 26 Nov 2025, 00:46, before the data leakage)**

In response to their concerns on scope of generalization, we clarified why our primary experiments focus on Stable Diffusion (the standard open-source T2I backbone used in prior backdoor works such as BadT2I and Rickrolling), and explicitly acknowledged the limitation to open-source models. We then extended our experiments to (i) SD 2.1 with different CLIP-based encoders (OpenAI CLIP, Openjourney CLIP, LoRA-finetuned CLIP), and (ii) LDM with a BERT-style text encoder (LDMbert), adding new results that show our attack remains effective under this different textual encoder architecture (more details can be found in the revised version of Section 5.2).

On the defense side, we added new LoRA fine-tuning experiments where we fine-tune the combined backdoor encoder and backdoor diffusion model on clean data for different training steps. The results (0/500/1000 fine-tuning steps) show only a slight drop in SSIM (pre-set image attack) and ASR (specific style attack), indicating that fine-tuning does not significantly weaken our backdoor. We further explain this behavior from the perspective of LoRA’s low-rank residual updates and the resulting small impact on text embeddings. Besides, we also added a more detailed discussion of the defense of Fine-grained Prompt Screening (which did not provide official code), explaining that its text-embedding similarity mechanism is essentially equivalent to the embedding-based defense we already evaluate, and why it is not effective against our specially designed triggered embeddings (more details can be found in the revised version of Section 5.3).

Regarding the attack significance and threat model, we clarify that the main contribution of this paper is to propose a new attack vector for the supply chain of the customized T2I application scenario. It differs substantially from previous T2I backdoor attacks in attack design, threat model, and stealth mechanism. The combinational backdoor design offers greater stealthiness and more controllable triggering conditions than previous T2I backdoor attacks. The attacker can selectively attack specific model developer, without affecting non-targeted model developers, which also reduces the exposure risk of backdoor attack. This reveals a unique backdoor attack angle in customized T2I model scenario, which has important implications for AI security communities. After clarification, the reviewer notes that the paper “does point out a possible threat in the supply chain of open-sourced T2I models”, which gives the work “more acceptable significance,” and that the demonstration has become “more complete and rigorous”, leading him/her to adopt “a more positive attitude toward the work.”

Overall, we believe our rebuttal address Reviewer hpav’s main concerns, and leading him/her to increase his/her score (at the time of 26 Nov 2025, 00:46, before the data leakage).

---

### Author Response · Authors · 2025-11-30
**Summary of responses to reviewer VUYx's concerns**

**We would like to thank this reviewer VUYx for recognizing the value of our work, as well as for their constructive suggestions, which helped us strengthen the evaluation methodology and the completeness of our experiments.**

In response to the concern that our style-ASR relied on a ResNet-18 classifier trained by us (potentially introducing bias), we followed the reviewer’s suggestion and added a GPT-4o–as–a–judge evaluation for the “specific style” backdoor. This not only aligns the style evaluation with our real-world case study setting but also removes potential bias on a custom classifier.

For the secret image hiding application, we added an explicit experiment using the cat image in Appendix E as the hidden target. We now report quantitative similarity metrics between the original image and the backdoor-generated image, achieving SSIM = 0.9420 and LPIPS = 0.0476, demonstrating that the proposed scheme can indeed function as a valid secret image transmission mechanism. At the same time, we clearly state that this application is a positive byproduct rather than a core contribution.

Finally, for the pre-set image backdoor, we incorporated the reviewer’s suggestion to include LPIPS as an additional similarity metric. The new results confirm that the backdoor can generate target images that are not only visually similar (via SSIM) but also perceptually close under LPIPS metric.

Overall, these changes directly address the reviewer’s concerns, and improve the clarity and rigor of the paper. We are grateful for reviewer’s positive evaluation and constructive suggestions that helped us improve the manuscript.

---

### Author Response · Authors · 2025-11-30
**Summary of responses to reviewer rcc7's concerns**

**We thank the reviewer rcc7 for his/her feedback, which helps us to substantially clarify and strengthen the paper. We have done our best to address his/her main concerns.**

On W1 (core contribution / threat model). We clarified that our work is not merely “tampering with multiple module” but is explicitly motivated by mainstream custom T2I workflows (HuggingFace diffusers, CivitAI, SD WebUI), where text encoders and conditional diffusion models are independently trained, distributed, and recombined. Within this realistic scenario (acknowledged by reviewer rcc7 “where the proposed threat model indeed exists”), we identify a new supply-chain attack vector for customized T2I pipelines, and introduce a new triggering mechanism where a specially designed triggered text embedding that acts as a bridge between a backdoored encoder and a backdoored diffusion model. It opens a new perspective on supplychain attacks in custom T2I model scenarios. The attack design enables “each component benign alone, malicious only in combination”, and supports developer-selective attacks: only developers who adopt a particular pair of text encoder and diffusion model are affected, while using either component alone remains safe. This threat model and contribution were acknowledged by the other two reviewers. Reviewer hpav said “does point out a possible threat in the supply chain of open-sourced T2I models”. Reviewer VUYx said “New attack surface”.

On W2 (methodology). We acknowledged that the underlying loss function forms are similar to previous work, because all T2I backdoor attacks share the same standard training backbone (encoder fine-tuning via embedding loss, and DDPM noise-prediction loss). It should be noted that what matters most about a loss function is its optimization objective, not merely that “the formula looks somewhat similar”. As we explained in our previous reply, our loss function optimization objective differs from that of prior works. Besides, we clarified that our contribution and novelty does not lie in proposing a brand-new mathematical loss, but in: how the objectives are decomposed across modules, how the special triggered embedding is used to connect encoder and diffusion model, and how this realizes a new attack mechanism/triggering paradigm.

On W3 (defense experiments). we explained that we focused on ONION, T2IShield, UFID, and embedding-similarity-based defenses because they have public implementations and are already used as standard baselines in prior T2I backdoor papers (Rickrolling, BAGM, BadT2I). For more recent method Fine-grained Prompt Screening (which did not provide official code), we added a conceptual and experimental discussion, explaining that its text-embedding similarity mechanism is essentially equivalent to the embedding-based defense we already evaluate, and why it is not effective against our specially designed triggered embeddings. Besides, we also added new LoRA fine-tuning experiments where we fine-tune the combined backdoor encoder and backdoor diffusion model on clean data for different training steps (more details can be found in the revised version of Section 5.3). We also clarified our “stealthiness” from two perspective: combinational stealth (each component avoid detection when inspected alone; the backdoor exists only backdoor components are combined together), and embedding-level stealth (triggered embeddings remain very close to normal embeddings).

On narrow attack coverage, we clarified that this as a deliberate design choice rather than a limitation: our attack introduce model-developer-selective, supply-chain compromises (in fact, by selecting the target backdoor component, the attacker can arbitrarily expand the attack surface). It is more aimed at controllable triggering than at a higher trigger probability or maximize attack coverage. It ensures the attacker has fine-grained control over when the backdoor activates, greatly reducing the chance that benign users will trigger it unintentionally. This is also more in line with the attack objectives of real-world backdoor attacks, which prioritize concealment, long-term latency, and controllable triggering.

We believe that these clarifications and revisions show that we have engaged with their concerns and have significantly strengthened the clarity of our threat model, the methodological, and the defense experiments part.

---

### Meta-Review · Area_Chair_3eQY · 2026-01-01

**Summary:**

This paper studies combinational backdoor attacks against customized Text-to-Image (T2I) models, motivated by the increasingly common practice of assembling T2I pipelines from independently fine-tuned text encoders and diffusion models. The authors propose a combinational backdoor design in which malicious behavior emerges only when a specific poisoned encoder and poisoned diffusion model are used together, aiming to improve stealthiness and controllability in supply-chain attack scenarios.

The reviewers’ opinions are mixed. One reviewer finds the work clearly presented, practically motivated, and experimentally thorough, and views the identification of a new attack surface in modular T2I pipelines as a meaningful contribution. Another reviewer becomes more positive after the rebuttal, acknowledging that the paper does highlight a realistic supply-chain threat. However, a third reviewer remains strongly negative, raising persistent concerns about novelty, methodological distinctiveness, and sufficiency of evidence.

From an Area Chair perspective, the rebuttal successfully clarifies the intended threat model and strengthens several aspects of the empirical evaluation, addressing a number of reviewer concerns and improving the paper relative to the initial submission. However, some core weaknesses remain. First, regarding novelty, while the authors convincingly motivate a new attack vector in customized T2I supply chains, the proposed method does not appear sufficiently novel at the methodological level. The attack largely relies on existing loss formulations and training pipelines used in prior text-to-image backdoor work, with the main distinction being how these objectives are applied across modular components. Although this is a reasonable design choice, it remains debatable whether it constitutes a fundamentally new attack mechanism, as opposed to a compositional extension of established techniques. Second, regarding evaluation, while the experiments demonstrate that the attack is effective, the evidence that it is meaningfully more stealthy or impactful than existing backdoor methods is limited. As a result, this raised the reviewer's concern about whether the empirical results fully substantiate the claimed advantages over prior approaches.

In light of the above comments and discussion, I have to reject this submission based on its current form.

**Reviewer Concerns:**

Addressed by the rebuttal:
(1) The rebuttal substantially clarified the threat model and attack scenario.
(2) Several evaluation-related issues were partially addressed.

Still outstanding:
(1) Despite the clarified attack vector, the core methodology relies heavily on existing loss formulations and training pipelines from prior T2I backdoor work, and it remains unclear whether the approach represents a fundamentally new attack mechanism rather than a compositional extension of known techniques.
(2) Strength of the empirical claims remains limited.

**Reviewer Scores:**

Reviewer hpav: I believe the score would rise to 6.

Reviewer VUYx: This reviewer was positive throughout and rated the paper 8 (accept).

Reviewer rcc7: This reviewer maintained a strongly critical stance and reiterated concerns about methodological novelty and insufficient differentiation from prior work even after the rebuttal. I believe that, even with full participation in the discussion, the score would remain at 2 (reject), as the rebuttal did not fully resolve the core concerns.

---

### Decision · Program_Chairs · 2026-01-26

Reject